# Fixed Confidence Best Arm Identification in the Bayesian Setting

**Kyoungseok Jang**
Universitá degli Studi di Milano
ksajks@gmail.com

**Junpei Komiyama**
New York University / RIKEN AIP
junpei@komiyama.info

**Kazutoshi Yamazaki**
The University of Queensland
k.yamazaki@uq.edu.au

## Abstract

We consider the fixed-confidence best arm identification (FC-BAI) problem in the Bayesian setting. This problem aims to find the arm of the largest mean with a fixed confidence level when the bandit model has been sampled from the known prior. Most studies on the FC-BAI problem have been conducted in the frequentist setting, where the bandit model is predetermined before the game starts. We show that the traditional FC-BAI algorithms studied in the frequentist setting, such as track-and-stop and top-two algorithms, result in arbitrarily suboptimal performances in the Bayesian setting. We also obtain a lower bound of the expected number of samples in the Bayesian setting and introduce a variant of successive elimination that has a matching performance with the lower bound up to a logarithmic factor. Simulations verify the theoretical results.

## 1 Introduction

In many sequential decision-making problems, the learner repeatedly chooses an arm (option) to play with and observes a reward drawn from the unknown distribution of the corresponding arm. One of the most widely-studied instances of such problems is the multi-armed bandit problem [Thompson, 1933, Robbins, 1952, Lai, 1987], where the goal is to maximize the sum of rewards during the rounds. Since the learner does not know the distribution of rewards, they need to explore the different arms, and yet, exploit the arms of the most rewarding arms so far. Different from the classical bandit formulation, there are situations where one is more interested in collecting information rather than maximizing intermediate rewards. The best arm identification (BAI) is a sequential decision-making problem in which the learner is only interested in identifying the arm with the highest mean reward. While the origin of this problem dates back to at least the 1950s [Bechhofer, 1954, Paulson, 1964, Gupta, 1977], recent work in the field of machine learning reformulated the problem [Audibert et al., 2010]. In the BAI, the learner needs to pull arms efficiently for better identification. To achieve efficiency and accuracy, the learner should determine which arm to choose based on the history, when to stop the sampling, and which arm to recommend as the learner's final decision.

There are two types of BAI problems depending on the optimization objective. In the fixed-budget (FB) setting [Audibert et al., 2010], the learner attempts to minimize the probability of error (misidentification of the best arm) given a limited number of arm pulls $T$. In the fixed confidence (FC) setting [Jamieson and Nowak, 2014], the learner attempts to minimize the number of arm pulls, subject to a predefined probability of error $\delta \in (0, 1)$. In this paper, we shall focus on the FC setting, which is useful when we desire a rigorous statistical guarantee.

38th Conference on Neural Information Processing Systems (NeurIPS 2024).

Most of the previous BAI studies focus on the frequentist setting, where the bandit model is chosen adversarially from some hypothesis class beforehand. In this setting, several algorithms, such as Track and Stop [Kaufmann et al., 2016a] and Top-two algorithms [Russo, 2016, Qin et al., 2017b, Jourdan et al., 2022], are widely known. These algorithms have an optimal sample complexity, meaning that they are one of the most sample-efficient algorithms among the class of $\delta$-correct algorithms.

The sample complexity of these algorithms is problem-dependent. To see this, consider the following example.

**Example 1.** (A/B/C testing) Consider A/B/C testing of web designs. We have three arms (web designs) from which we would like to find the largest retention rate via allocating users to web designs $i = 1, 2, 3$. If we attempt to find the best arm with confidence $\delta$, we may need a large number of samples (users) when the suboptimality gap (the gap between the retention rate of the best arm and the second best arm) is small because in such a case the identification of the best arm is difficult – the minimum number of samples required is inversely proportional to the square of the suboptimality gap[Kaufmann et al., 2014]. For example, when comparing the testing of retention rates of (0.9, 0.5, 0.1) with (0.9, 0.89, 0.1), the second case requires around $\left(\frac{0.9-0.5}{0.9-0.89}\right)^2 = 1600$ times more samples compared to the first case.

In practice, the retention rate of $0.89$ in the second case may be acceptably good compared to the optimal retention rate of $0.9$, and we may stop exploration at the moment the learner identifies a reasonably good arm, which is the first or the second arm in this example. This idea is formalized in several ways. The literature of Ranking and Selection (R&S) usually considers the indifference-zone formulation [Hong et al., 2021]. In the context of best arm identification, a similar notion of $\epsilon$-best answer identification has also been considered [Maron and Moore, 1993, Even-Dar et al., 2006, Gabillon et al., 2012, Kaufmann and Kalyanakrishnan, 2013, Jourdan et al., 2023]. In these settings, the learner accepts a sub-optimal arm whose means are at most $\epsilon$ worse than the mean of the optimal arm. Other related settings include the good arm identification problem [Kano et al., 2019, Tabata et al., 2020, Zhao et al., 2023], where the goal is to identify an arm that exceeds the predefined threshold, and the thresholding bandit problem [Locatelli et al., 2016, Xu et al., 2019], where the goal is to identify whether the mean of each arm is above or below the threshold. All these problem settings require an extra parameter, like $\epsilon$ or an acceptance threshold, that directly determines the acceptance level. Even though the algorithm's performance depends on this parameter, it is often challenging to determine a reasonable value for it in advance.

In this paper, we study an alternative approach based on the Bayesian setting. In particular, we consider the prior distribution on the model parameters. We relax the requirement on the correctness of the best arm identification by using the prior belief. Rather than requiring the frequentist $\delta$-correctness for any model, we require the learner to have marginalized correctness over the prior distribution, which we call Bayesian $\delta$-correctness.

We study the fixed confidence BAI (FC-BAI) problem in the Bayesian setting. Our contributions are as follows.

- **First,** we find that in the Bayesian setting, the performance of the traditional frequentist setting-based algorithms, such as Track and Stop and Top-two algorithms, can be arbitrarily worse (Section 3). This is because frequentist approaches spend too many resources when the suboptimality gap is narrow.

- **Second,** we prove that the lower bound of the number of expected samples should attain at least the order of $\Omega(\frac{L(\boldsymbol{H})^2}{\delta})$ as $\delta \to 0$ (Section 4). Here $L(\boldsymbol{H})$ is our novel quantity that represents the sample complexity with respect to the prior distribution $\boldsymbol{H}$. This order is different from the existing lower bound in the frequentist setting[1], implying that the Bayesian setting is essentially different from the frequentist setting.

- **Third,** we design an algorithm whose expected sample size is upper-bounded by $O(\frac{L(\boldsymbol{H})^2}{\delta} \log \frac{L(\boldsymbol{H})}{\delta})$ (Section 5). Our algorithm is based on the elimination algorithm [Maron and Moore, 1993, Even-Dar et al., 2006, Frazier, 2014], but we add an early stopping criterion to prevent over-commitment of the algorithm for a bandit model with a narrow suboptimality gap. Our algorithm has a matching upper bound up to the logarithmic factor.

---

[1]In fact, marginalizing the frequentist sample complexity over the prior distribution leads to an unbounded value.

We also conduct simulation to demonstrate that the sample complexity of frequentist algorithms does indeed diverge in a bandit model with a small suboptimality gap, even in very simple cases (Section 6).

## 1.1 Related work

To our knowledge, BAI problems studied for the Bayesian setting have been limited to the fixed budget setting [Komiyama et al., 2023, Atsidakou et al., 2023]. Komiyama et al. [2023] showed that, in the fixed-budget setting, a simple non-Bayesian algorithm has an optimal simple regret up to a constant factor, implying that the advantage the learner could get from the prior is small when the budget is large. This is very different from our fixed-confidence setting, where utilizing the prior distribution is necessary.

Several FC-BAI algorithms used Bayesian ideas on the structure of the algorithm, although most of those studies used frequentist settings for measuring the guarantee. The 'Top-Two' type of algorithms are the leading representatives in this direction. The first instance of top-two algorithms, which is called Top-Two Thompson sampling (TTTS), is introduced in the context of Bayesian best arm identification. TTTS requires a prior distribution, and Russo [2016] showed that the sample complexity of posterior convergence of TTTS which is the same as the sample complexity of the frequentist fixed-confidence best arm identification. Subsequent research analyzed the performance of TTTS from the frequentists' viewpoint [Shang et al., 2020]. Later on, the idea of top-two sampling is then extended into many other algorithms, such as Top-Two Transportation Cost [Shang et al., 2020], Top-Two Expected Improvement (TTEI, Qin et al. 2017b), Top-Two Upper Confidence bound (TTUCB, Jourdan and Degenne 2022a). Even though some of the top two algorithms adapt a prior, they implicitly solve the optimization that is justified in view of frequentist.

Another line of Bayesian sequential decision-making is Bayesian optimization [Srinivas et al., 2010, Mockus, 2012, Shahriari et al., 2016, Jamieson and Talwalkar, 2016, Frazier, 2018], where the goal is to find the best arm in Bayesian setting. Note that Bayesian optimization tends to deal with structured identification, especially for Gaussian processes, and most of the algorithms for Bayesian optimization do not have specific stopping criteria.

## 2 Problem setup

We study the fixed confidence best arm identification problem (FC-BAI) in a Bayesian setting. In this setup, we have $k$ arms in the set $[k] := \{1, 2, \ldots, k\}$ with *unknown* distribution $\boldsymbol{P} = (P_1, \cdots, P_k)$ which is drawn from a *known* prior distribution at time 0, namely $\boldsymbol{H} = (H_1, \cdots, H_k)$. The unknown bandit model $P_i$ is a one-parameter distribution, and $\boldsymbol{P}$ is specified by $\boldsymbol{\mu} := (\mu_1, \cdots, \mu_k)$. To simplify the problem, we will focus on the Gaussian case, where each $P_i$ is a Gaussian distribution with known variance $\sigma_i^2$. Each mean of $P_i$, denoted $\mu_i$, is drawn from a known prior Gaussian distribution $H_i$, which can be written as $N(m_i, \xi_i^2)$.

At every time step $t = 1, 2, \cdots$, the forecaster chooses an arm $A_t \in [k]$ and observes a reward $X_t$, which is drawn independently from $P_{A_t}$. Since we focus on the Gaussian case, $X_t \sim N(\mu_{A_t}, \sigma_{A_t}^2)$ conditionally given $A_t$ and $\mu_{A_t}$. After each sampling, the forecaster must decide whether to continue the sampling process or stop sampling and make a recommendation $J \in [k]$.

Let $\mathcal{F}_t = \sigma(A_1, X_1, A_2, X_2, \cdots, A_t, X_t)$ be the $\sigma$-field generated by observations up to time $t$. The algorithm of the forecaster $\pi := ((A_t)_t, \tau, J)$ is defined by the following triplet [Kaufmann et al., 2016a]:

- A sampling rule $(A_t)_t$, which determines the arm to draw at round $t$ based on the previous history (each $A_t$ must be $\mathcal{F}_{t-1}$ measurable).

- A stopping rule $\tau$, which means when to stop the sampling (i.e., stopping time with respect to $\mathcal{F}_t$).

- A decision rule $J$, which determines the arm the forecaster recommends based on his sampling history (i.e., $J$ is $\mathcal{F}_\tau$-measurable).

In FC-BAI, the forecaster aims to recommend arm $J$ that correctly identifies (one of) the best arm(s) $i^*(\boldsymbol{\mu}) := \arg\max_{i \in [k]} \mu_i$ with probability at least $1 - \delta$. Since the case of multiple best arms is of

measure zero under $\boldsymbol{H}$, we can focus on $\boldsymbol{\mu}$ such that $i^*(\boldsymbol{\mu})$ is unique. For the FC-BAI problem in the Bayesian setting, we use the *expected* probability of misidentification:

$$\text{PoE}(\pi; \boldsymbol{H}) := \mathbb{E}_{\boldsymbol{\mu} \sim \boldsymbol{H}}\left[\mathbb{P}\left(J \neq i^*(\boldsymbol{\mu})|\mathcal{H}_{\boldsymbol{\mu}}\right)\right], \tag{1}$$

where $\mathcal{H}_{\boldsymbol{\mu}} := \{\boldsymbol{\mu} \text{ is the correct bandit model}\}$. Now we formally define the algorithm of interest as follows:

**Definition 1.** (Bayesian $\delta$-correctness) For a prior distribution $\boldsymbol{H}$, an algorithm $\pi = ((A_t), \tau, J)$ is said to be Bayesian $(\boldsymbol{H}, \delta)$-correct if it satisfies $\text{PoE}(\pi; \boldsymbol{H}) \leq \delta$. Let $\mathcal{A}^b(\delta, \boldsymbol{H})$ be the set of Bayesian $(\boldsymbol{H}, \delta)$-correct algorithms for the prior distribution $\boldsymbol{H}$.

The objective of the FC-BAI problem in the Bayesian setting is to find an algorithm $\pi = ((A_t)_t, \tau, J) \in \mathcal{A}^b(\delta, \boldsymbol{H})$ that minimizes $\mathbb{E}_{\boldsymbol{\mu} \sim \boldsymbol{H}}[\tau]$.

**Terminology** Define $N_i(t) = \sum_{s=1}^{t-1} \mathbf{1}[A_s = i]$ as the number of times arm $i$ is pulled before timestep $t$. Let $h_i$ be the probability density function of $H_i$. Since we consider Gaussian prior, $h_i(\mu_i) := (1/\sqrt{2\pi}\xi_i) \exp(-(\mu_i - m_i)^2/(2\xi_i^2))$. Let $i^*, j^* : \mathbb{R}^k \to [k]$ be the best and the second best arm under the input such that for each $\boldsymbol{\mu} \in \{\mathbf{x} \in \mathbb{R}^k : x_i \neq x_j \forall i \neq j\}$, $i^*(\boldsymbol{\mu}) = \arg\max_{i \in [K]} \mu_i$ and $j^*(\boldsymbol{\mu}) = \arg\max_{i \in [K] \setminus \{i^*(\boldsymbol{\mu})\}} \mu_i$.

Let $\text{KL}_i(a\|b) := \frac{(a-b)^2}{2\sigma_i^2}$ represent the KL-divergence between two Gaussian distributions with equal variances (the variance of the $i$-th arm $\sigma_i^2$) but different means, denoted as $a$ and $b$. Similarly, $d(a, b) := a \log(a/b) + (1-a) \log((1-a)/(1-b))$ is the KL divergence between two Bernoulli distributions with means $a$ and $b$. Throughout this paper, $\mathbb{E}_{\boldsymbol{\mu}}$ and $\mathbb{P}_{\boldsymbol{\mu}}$ denote the expectation and probability when the bandit model is fixed as $\boldsymbol{\mu} \in \mathbb{R}^k$, i.e., $\mathbb{E}_{\boldsymbol{\mu}} = \mathbb{E}[\cdot|\mathcal{H}_{\boldsymbol{\mu}}]$ and $\mathbb{P}_{\boldsymbol{\mu}} = \mathbb{P}(\cdot|\mathcal{H}_{\boldsymbol{\mu}})$. We will abuse the notation PoE so that for $\boldsymbol{\lambda} \in \mathbb{R}^k$, $\text{PoE}(\pi; \boldsymbol{\lambda})$ means

$$\text{PoE}(\pi; \boldsymbol{\lambda}) := \mathbb{P}_{\boldsymbol{\lambda}}\left(J \neq i^*(\boldsymbol{\lambda})|\mathcal{H}_{\boldsymbol{\lambda}}\right).$$

Naturally, $\text{PoE}(\pi; \boldsymbol{H}) = \mathbb{E}_{\boldsymbol{\mu} \sim \boldsymbol{H}}\left[\text{PoE}(\pi; \boldsymbol{\mu})\right]$.

Lastly, we introduce the constant $L(\boldsymbol{H})$ that characterizes the sample complexity in the Bayesian setting.

**Definition 2.** For each $i, j \in [k]$, define $L_{ij}(\boldsymbol{H})$ and $L(\boldsymbol{H})$ as follows:

$$L(\boldsymbol{H}) := \sum_{i,j \in [k], i \neq j} L_{ij}(\boldsymbol{H}) \text{ where } L_{ij}(\boldsymbol{H}) := \int_{-\infty}^{\infty} h_i(x) h_j(x) \prod_{s: s \in [k] \setminus \{i,j\}} H_s(x) \, \mathrm{d}x.$$

This constant has the following interesting property which we call a volume lemma:

**Lemma 1** (Volume Lemma, informal). For $\Delta \in (0, 1)$, let

$$L(\boldsymbol{H}, \Delta) := \frac{1}{\Delta} \mathbb{P}_{\boldsymbol{\mu} \sim \boldsymbol{H}}\left[\mu_{i^*(\boldsymbol{\mu})} - \mu_{j^*(\boldsymbol{\mu})} \leq \Delta\right].$$

Then, $\lim_{\Delta \to 0^+} L(\boldsymbol{H}, \Delta) = L(\boldsymbol{H})$. In particular, for $\Delta < \frac{L(H)}{\sum_{i \in [k]} \frac{2(k-1)}{\xi_i}}$, $L(H, \Delta) \in (\frac{1}{2} L(H), 2L(H))$.

The volume lemma states that the volume of prior where the suboptimality gap is smaller than $\Delta$ is proportional to $L(\boldsymbol{H})\Delta$ when $\Delta$ is small. We will see in Section 3 that such small-gap cases, which require a large amount of exploration to identify the best arm, dominate the Bayesian expectation of the stopping time. Therefore, $L(\boldsymbol{H})$ defines the Bayesian sample complexity. The formal version of this lemma, which involves some regularity conditions, is shown in Appendix B.

**Remark 1.** Here, we elaborate on how the Bayesian sample complexity is defined. As will be shown in Section 3, for an algorithm to have a finite expected stopping time, it must determine whether the current instance is difficult or not. In particular, if an algorithm tries to identify even the top-$O(\delta)$ 'hardest instances'[2] in the prior, the algorithm cannot achieve the finite expected stopping time. By

---

[2]Here the hardness is based on the size of the suboptimality gap. When the suboptimality gap of an instance is small, it is a harder instance for the BAI algorithm to identify the best arm, as mentioned in Example 1.

Lemma 1, the suboptimality gap of the top-$O(\delta)$ hardest instance is given by $L(\mathcal{H})\Delta \approx \delta$, and the corresponding (frequentist) sample complexity is proportional to $\Delta^{-2} = \left(L(\mathcal{H})/\delta\right)^2$ [Kaufmann et al., 2014]. Such instances constitute an $O(\delta)$ fraction of the prior, and thus the Bayesian sample complexity is:

$$O\left(\frac{(L(\mathcal{H}))^2}{\delta^2} \times \delta\right) = O\left(\frac{(L(\mathcal{H}))^2}{\delta}\right).$$

## 3 Limitation of traditional frequentist approaches in the Bayesian setting

Existing BAI studies mainly focused on the Frequentist $\delta$-correct algorithms which are defined as follows:

**Definition 3** (Frequentist $\delta$-correctness). An algorithm $\pi = ((A_t), \tau, J)$ is said to be frequentist $\delta$-correct if, for any bandit instance $\boldsymbol{\mu} \in \mathbb{R}^k$ such that $i^*(\boldsymbol{\mu})$ is unique, it satisfies $\mathrm{PoE}(\pi; \boldsymbol{\mu}) \leq \delta$. Let $\mathcal{A}^f(\delta)$ be the set of all frequentist-$\delta$-correct algorithms.

For the frequentist $\delta$-correct algorithms, Garivier and Kaufmann [2016] proved a lower bound for the expected stopping time as follows: for all bandit instance $\boldsymbol{\mu} \in \mathbb{R}^k$ and for all $((A_t), \tau, J) \in \mathcal{A}^f(\delta)$,

$$\mathbb{E}_{\boldsymbol{\mu}}[\tau] \geq \log(\delta^{-1})T^*(\boldsymbol{\mu}) + o(\log(\delta^{-1})) \tag{2}$$

where $T^*(\boldsymbol{\mu})$ is a sample complexity function dependent on the bandit instance $\boldsymbol{\mu}$.[3] Moreover, many of the known frequentist $\delta$-correct algorithms achieve asymptotic optimality [Garivier and Kaufmann, 2016, Russo, 2016, Tabata et al., 2023, Qin et al., 2017a], meaning that they are orderwisely tight up to the lower bound on Eq. (2) as $\delta \to 0$. However, little is known, or at least discussed, about their performance in the Bayesian setting.

One can check that a frequentist $\delta$-correct algorithm is also Bayesian $\delta$-correct as well ($\mathcal{A}^f(\delta) \subset \mathcal{A}^b(\delta, \boldsymbol{H})$ for all $\boldsymbol{H}$). Naturally, our interest is whether or not the most efficient classes of frequentist $\delta$-correct algorithms, such as Tracking algorithms and Top-two algorithms, are efficient in Bayesian settings. Somewhat surprisingly, the following theorem states that any $\delta$-correct algorithm is suboptimal in Bayesian settings.

**Theorem 2.** For all $\delta > 0$, $\boldsymbol{H}$ and $((A_t), \tau, J) \in \mathcal{A}^f(\delta)$, $\mathbb{E}_{\boldsymbol{\mu} \sim \boldsymbol{H}}[\tau] = +\infty$.

Proof of Theorem 2 is found in Appendix C. To illustrate the proof, we will use a two-armed Gaussian instance as an example.

### 3.1 Special case - two armed Gaussian case

Here we present one intuitive corollary of the lower bound theorem [Kaufmann et al., 2016a, Garivier and Kaufmann, 2016, Kaufmann et al., 2016b] that uses a standard information-theoretic technique.

**Corollary 3** (Kaufmann et al. 2014). Let $\delta \in (0,1)$. For any frequentist $\delta$-correct algorithm $((A_t), \tau, J)$ and for any fixed mean vector $\boldsymbol{\mu} = (\mu_1, \mu_2) \in \mathbb{R}^2$, $\mathbb{E}_{\boldsymbol{\mu}}[\tau] \geq \frac{d(\delta, 1-\delta)}{(\mu_1 - \mu_2)^2} \geq \frac{\log \frac{1}{2.4\delta}}{(\mu_1 - \mu_2)^2}$.

In the frequentist setting, Corollary 3 implies the lower bound of $\mathbb{E}_{\boldsymbol{\mu}}[\tau] = \Omega(\log(\delta^{-1})/(\mu_1 - \mu_2)^2)$, which is $\Omega(\log(\delta^{-1}))$ when we view parameters $(\mu_1, \mu_2)$ as constants. However, in the Bayesian setting, the algorithm is given the prior distribution $\boldsymbol{H}$ on $\boldsymbol{\mu}$, and thus the stopping time is marginalized over $\boldsymbol{H}$. In particular, limiting our interest to the case of $|\mu_1 - \mu_2| < \Delta$ for small enough $\Delta > 0$, we can obtain the following lower bound:

$$
\begin{aligned}
\mathbb{E}_{\boldsymbol{\mu} \sim \boldsymbol{H}}[\tau] &\geq \mathbb{E}_{\boldsymbol{\mu} \sim \boldsymbol{H}}[\tau \cdot \mathbf{1}[|\mu_1 - \mu_2| \leq \Delta]] &&\text{(Since } \tau \text{ is positive r.v.)} \\
&\geq \mathbb{E}_{\boldsymbol{\mu} \sim \boldsymbol{H}}\left[\mathbb{E}[\tau|\boldsymbol{\mu}] \cdot \mathbf{1}[|\mu_1 - \mu_2| \leq \Delta]\right] &&\text{(Law of total expectation)} \\
&\geq \mathbb{E}_{\boldsymbol{\mu} \sim \boldsymbol{H}}\left[\frac{\log \delta^{-1}}{(\mu_1 - \mu_2)^2} \cdot \mathbf{1}[|\mu_1 - \mu_2| \leq \Delta]\right] &&\text{(Corollary 3)} \\
&\geq \frac{\log \delta^{-1}}{\Delta^2} \mathbb{P}_{\boldsymbol{\mu} \sim \boldsymbol{H}}[|\mu_1 - \mu_2| \leq \Delta] \geq \frac{\log \delta^{-1}}{\Delta^2} \frac{L(\boldsymbol{H})}{2} \Delta &&\text{(Lemma 1)}
\end{aligned}
$$

---

[3]For details about $T^*(\boldsymbol{\mu})$, a reader may refer to Garivier and Kaufmann [2016].

$$= \Omega \left( \frac{L(\boldsymbol{H}) \log \delta^{-1}}{\Delta} \right).$$

This inequality implies that if we naively use a known frequentist $\delta$-correct algorithm in the Bayesian setting, the expected stopping time will diverge because we can choose an arbitrarily small $\Delta$. The case of a small gap is *difficult to identify*, and the expected stopping time can be very large for such a case if we aim to identify the best arm for any model.

# 4  Lower bound

This section will elaborate on the lower bound of the stopping time in the Bayesian setting. Theorem 4 below states that any Bayesian $(\boldsymbol{H}, \delta)$-correct algorithm requires the expected stopping time of at least $\Omega(\frac{L(\boldsymbol{H})^2}{\delta})$.

**Theorem 4.** Define $\sigma_{\min} = \min_{i \in [k]} \sigma_i^2$ and $N_V = \frac{L(\boldsymbol{H})^2 \sigma_{\min}^2 \ln 2}{16 e^4 \delta}$. Let $\delta < \delta_L(\boldsymbol{H})$ be sufficiently small.[4] Then, for any BAI algorithms $\pi = ((A_t), \tau, J)$, if $\mathbb{E}_{\boldsymbol{\mu} \sim \boldsymbol{H}}[\tau] \leq N_V$, then $\mathrm{PoE}(\pi; \boldsymbol{H}) \geq \delta$.

In this main body, we will use the two-armed Gaussian bandit model with homogeneous variance condition (i.e. $\sigma_1 = \sigma_2 = \sigma$) for easier demonstration of the proof sketch. Theorem 4, which is more general in the sense that it can deal with $k > 2$ arms with heterogeneous variances, is proven in Appendix D.

**Sketch of the proof, for $k = 2$:**  It suffices to show that the following is an empty set:

$$\mathcal{A}^b(\delta, \boldsymbol{H}, N_V) := \{\pi \in \mathcal{A}^b(\delta, \boldsymbol{H}) : \mathbb{E}_{\boldsymbol{\mu} \sim \boldsymbol{H}}[\tau] \leq N_V\}.$$

Assume that $\mathcal{A}^b(\delta, \boldsymbol{H}, N_V) \neq \emptyset$ and choose an arbitrary $\pi \in \mathcal{A}^b(\delta, \boldsymbol{H}, N_V)$. We start from the following transportation lemma:

**Lemma 5** (Kaufmann et al. 2016a, Lemma 1)**.** Let $\delta \in (0, 1)$. For any algorithm $((A_t), \tau, J)$, any $\mathcal{F}_\tau$-measurable event $\mathcal{E}$, any bandit models $\boldsymbol{\mu}, \boldsymbol{\lambda} \in \{(x, y) \in \mathbb{R}^2 : x \neq y\}$ such that $i^*(\boldsymbol{\mu}) \neq i^*(\boldsymbol{\lambda})$,

$$\mathbb{E}_{\boldsymbol{\mu}} \left[ \sum_{i=1}^2 \mathrm{KL}_i(\mu_i, \lambda_i) N_i(\tau) \right] \geq d(\mathbb{P}_{\boldsymbol{\mu}}(\mathcal{E}), \mathbb{P}_{\boldsymbol{\lambda}}(\mathcal{E})).$$

Note that the above Lemma holds for any algorithm, and thus works for any stopping time $\tau$. Now define $\boldsymbol{\nu}(\boldsymbol{\mu})$ as a swapped version of $\boldsymbol{\mu} \in \mathbb{R}^2$, which means $(\boldsymbol{\nu}(\boldsymbol{\mu}))_1 = \mu_2, \boldsymbol{\nu}(\boldsymbol{\mu})_2 = \mu_1$, and let $\mathcal{E} = \{J \neq i^*(\boldsymbol{\mu})\}$, the event that the recommendation of the algorithm is wrong. Substituting $\boldsymbol{\lambda}$ with $\boldsymbol{\nu}(\boldsymbol{\mu})$ from the above equation of Lemma 5 leads to

$$\mathbb{E}_{\boldsymbol{\mu}} \left[ \frac{(\mu_1 - \mu_2)^2}{2\sigma^2} \tau \right] \geq d(\mathrm{PoE}(\pi; \boldsymbol{\mu}), 1 - \mathrm{PoE}(\pi; \boldsymbol{\nu})) \geq \log \frac{2}{2.4(\mathrm{PoE}(\pi; \boldsymbol{\mu}) + \mathrm{PoE}(\pi; \boldsymbol{\nu}))}. \quad (3)$$

Note that the first inequality comes from the fact that $\mathcal{E}$, the failure event of the bandit model $\boldsymbol{\mu}$, is exactly a success event of $\boldsymbol{\nu}(\boldsymbol{\mu})$ in this two-armed case, and the last inequality is from our modified lemma (Lemma 11) from Eq. (3) of Kaufmann et al. [2016a]. One can rewrite the above inequality as

$$\frac{\mathrm{PoE}(\pi; \boldsymbol{\mu}) + \mathrm{PoE}(\pi; \boldsymbol{\nu})}{2} \geq \frac{1}{2.4} \exp \left( \mathbb{E}_{\boldsymbol{\mu}} \left[ -\frac{(\mu_1 - \mu_2)^2}{2\sigma^2} \tau \right] \right). \quad (4)$$

We can rewrite the conditions of $\mathcal{A}^b(\delta, \boldsymbol{H}, N_V)$ as

$$\mathrm{PoE}(\pi; \boldsymbol{H}) = \int_{\boldsymbol{\mu} \in \mathbb{R}^2} \mathrm{PoE}(\pi; \boldsymbol{\mu}) \, d\boldsymbol{H}(\boldsymbol{\mu}) \leq \delta \quad \text{and} \quad \int_{\boldsymbol{\mu} \in \mathbb{R}^2} \mathbb{E}_{\boldsymbol{\mu}}[\tau] \, d\boldsymbol{H}(\boldsymbol{\mu}) \leq N_V. \quad (\text{Opt0})$$

Using Eq. (4) and with some symmetry tricks, we get $V_0 \leq \mathrm{PoE}(\pi; \boldsymbol{H})$ where

$$V_0 := \int_{\boldsymbol{\mu} \in \mathbb{R}^2} \frac{1}{2.4} \exp \left( -\frac{(\mu_1 - \mu_2)^2}{2\sigma^2} \mathbb{E}_{\boldsymbol{\mu}}[\tau] \right) d\boldsymbol{H}(\boldsymbol{\mu}) \leq \delta \quad \text{and} \quad \int_{\boldsymbol{\mu} \in \mathbb{R}^2} \mathbb{E}_{\boldsymbol{\mu}}[\tau] \, d\boldsymbol{H}(\boldsymbol{\mu}) \leq N_V.$$
$$(\text{Opt1})$$

---

[4]In particular, $\delta_L(\boldsymbol{H})$ is defined in Appendix G.

---

**Algorithm 1** Successive Elimination with Early-Stopping

---

**Input:** Confidence level $\delta$, prior $\boldsymbol{H}$
$\Delta_0 := \frac{\delta}{4L(\boldsymbol{H})}$
Initialize the candidate of best arms $\mathcal{A}(1) = [K]$.
$t = 1$
**while** True **do**
   Draw each arm in $\mathcal{A}(t)$ once. $t \leftarrow t + |\mathcal{A}(t)|$.
   **for** $i \in \mathcal{A}(t)$ **do**
      Compute $\mathrm{UCB}(i,t)$ and $\mathrm{LCB}(i,t)$ from (5).
      **if** $\mathrm{UCB}(i,t) \leq \max_j \mathrm{LCB}(j,t)$ **then**
         $\mathcal{A}(t) \leftarrow \mathcal{A}(t) \setminus \{i\}$.
      **end if**
   **end for**
   **if** $|\mathcal{A}(t)| = 1$ **then**
      **Return** arm $J$ in $\mathcal{A}(t)$.
   **end if**
   Compute $\hat{\Delta}^{\mathrm{safe}}(t) := \max\limits_{i \in \mathcal{A}(t)} \mathrm{UCB}(i,t) - \max\limits_{i \in \mathcal{A}(t)} \mathrm{LCB}(i,t)$.
   **if** $\hat{\Delta}^{\mathrm{safe}}(t) \leq \Delta_0$ **then**
      **Return** arm $J$, uniformly sampled from $\mathcal{A}(t)$.
   **end if**
**end while**

---

Note that on the above two inequalities, only $\mathbb{E}_{\boldsymbol{\mu}}[\tau]$ is the value that depends on the algorithm $\pi$. Now our main idea is that we can relax these two inequalities to the following optimization problem by substituting $\mathbb{E}_{\boldsymbol{\mu}}[\tau]$ to an arbitrary $\tilde{n} : \mathbb{R}^2 \to [0, \infty)$ as follows:

$$V := \inf_{\tilde{n}:\mathbb{R}^2 \to [0,\infty)} \int_{\boldsymbol{\mu} \in \mathbb{R}^2} \exp\left(-\frac{(\mu_1 - \mu_2)^2}{2\sigma^2} \tilde{n}(\boldsymbol{\mu})\right) \mathrm{d}\boldsymbol{H}(\boldsymbol{\mu}) \text{ s.t. } \int_{\boldsymbol{\mu} \in \mathbb{R}^2} \tilde{n}(\boldsymbol{\mu}) \, \mathrm{d}\boldsymbol{H}(\boldsymbol{\mu}) \leq N_V$$
$$\text{(Opt2)}$$

and $V \leq 2.4 V_0 \leq 2.4\delta$. Let $\mathcal{N} := \{\boldsymbol{\mu} \in \mathbb{R}^2 : |\mu_1 - \mu_2| < \Delta := \frac{8\delta}{L(\boldsymbol{H})}\}$. From the discussions in Section 3.1, one might notice that $\mathcal{N}$ is an important region for bounding $\mathbb{E}_{\boldsymbol{\mu} \sim \boldsymbol{H}}[\tau]$. We can relax the above (Opt2) to the following version, which focuses more on $\mathcal{N}$:

$$V' := \inf_{\tilde{n}:\mathbb{R}^2 \to [0,\infty)} \int_{\boldsymbol{\mu} \in \mathcal{N}} \exp\left(-\frac{(\mu_1 - \mu_2)^2}{2\sigma^2} \tilde{n}(\boldsymbol{\mu})\right) \mathrm{d}\boldsymbol{H}(\boldsymbol{\mu}) \text{ s.t. } \int_{\boldsymbol{\mu} \in \mathcal{N}} \tilde{n}(\boldsymbol{\mu}) \, \mathrm{d}\boldsymbol{H}(\boldsymbol{\mu}) \leq N_V.$$
$$\text{(Opt3)}$$

One can prove $V' \leq V \leq 2.4\delta$. Now, if we notice that function $x \mapsto \exp\left(-\frac{(\mu_1 - \mu_2)^2}{2\sigma^2} x\right)$ is a convex function, we can use Jensen's inequality to verify that the optimal solution for (Opt3) is when $\tilde{n} = \frac{N_V}{\mathbb{E}_{\boldsymbol{\mu} \sim \boldsymbol{H}}[\mathbf{1}_{\mathcal{N}}]} \mathbf{1}_{\mathcal{N}}$, and when we use $N_V$ in Theorem 4, one can get:

$$V' \geq \int_{\boldsymbol{\mu} \in \mathcal{N}} \exp\left(-\frac{\Delta^2}{2\sigma^2} \cdot \left(\frac{N_V}{\mathbb{E}_{\boldsymbol{\mu} \sim \boldsymbol{H}}[\mathbf{1}_{\mathcal{N}}]}\right)\right) \mathrm{d}\boldsymbol{H}(\boldsymbol{\mu}) \qquad \text{(by optimality of } \tilde{n} = \tfrac{N_V}{\mathbb{E}_{\boldsymbol{\mu} \sim \boldsymbol{H}}[\mathbf{1}_{\mathcal{N}}]})$$

$$\geq \int_{\boldsymbol{\mu} \in \mathcal{N}} \exp\left(-\frac{\Delta^2 \cdot N_V}{\sigma^2 \Delta L(\boldsymbol{H})}\right) \mathrm{d}\boldsymbol{H}(\boldsymbol{\mu}) \geq \exp\left(-\frac{\Delta \cdot N_V}{\sigma^2 L(\boldsymbol{H})}\right) \cdot \Delta L(\boldsymbol{H}) \quad \text{(both by Lemma 1)}$$

$$> 2.4\delta, \qquad\qquad\qquad\qquad\qquad\qquad\qquad\qquad\qquad \text{(by definition of } N_V \text{ and } \Delta)$$

which is a contradiction. This means no algorithm satisfies (Opt1), and the proof is completed.

## 5 Main algorithm

This section introduces our main algorithm (Algorithm 1). In short, our algorithm is a modification of the elimination algorithm with the incorporation of the indifference zone technique. Define

$\Delta_0 := \frac{\delta}{4L(\boldsymbol{H})}$ which satisfies the following condition, thanks to Lemma 1:

$$\mathbb{P}_{\boldsymbol{\mu} \sim \boldsymbol{H}}(\mu_{i^*(\boldsymbol{\mu})} - \mu_{j^*(\boldsymbol{\mu})} \leq \Delta_0) \leq \frac{\delta}{2}.$$

In each iteration of the **while** loop of Algorithm 1, the learner selects and observes each arm in the active set. After drawing each arm once, the algorithm calculates the confidence bounds for each arm in the active set using the formula as follows: let $\mathrm{Conf}(i, t)$ and $\hat{\mu}_i(t)$ be the confidence width and the empirical mean of arm $i$ at time $t$ as

$$\mathrm{Conf}(i, t) := \sqrt{2\sigma_i^2 \frac{\log(6(N_i(t))^2/((\frac{\delta^2}{2K})\pi^2))}{N_i(t)}}, \qquad \hat{\mu}_i(t) := \sum_{s=1}^{t-1} X_s \mathbf{1}[A_s = i].$$

Then the upper and lower confidence bounds of arm $i$ at timestep $t$, denoted as UCB and LCB respectively, can be defined in the following manner:

$$\mathrm{UCB}(i, t) := \hat{\mu}_i(t) + \mathrm{Conf}(i, t), \qquad \mathrm{LCB}(i, t) := \hat{\mu}_i(t) - \mathrm{Conf}(i, t). \tag{5}$$

This confidence bounds ensure that, with high probability, for all $t \in [T]$ and $i \in [K]$, $\mu_i \in (\mathrm{UCB}(i, t), \mathrm{LCB}(i, t))$ (See Lemma 15 in Appendix for details). After calculating UCB and LCB, the algorithm eliminates arms with UCB smaller than the largest LCB and maintains only arms that could be optimal in the active set $\mathcal{A}$. Up to this point, it follows the traditional elimination approach.

The main difference in our algorithm lies in the stopping criterion. At the end of each iteration, the algorithm checks the stopping criterion. Unlike typical elimination algorithms that continue until only one arm remains, we have introduced an additional indifference condition. This condition arises when the suboptimality gap is so small that identifying them would require an excessive number of samples. In such cases, our algorithm stops additional attempts to identify differences between arms in the active set and randomly recommends one from the active set instead.

**Remark 2.** In the context of PAC-$(\epsilon, \delta)$ identification, Even-Dar et al. [2006, Remark 9] introduced a similar approach. The largest difference is that they use the parameter $\epsilon$ as a parameter that defines the indifference-zone level, whereas our parameter $\Delta_0$ is spontaneously derived from the prior $\boldsymbol{H}$ and the confidence level $\delta$ without specifying the indifference-zone.

Theorem 6 describes the theoretical guarantee of the Algorithm 1.

**Theorem 6.** For $\delta < 4L(\boldsymbol{H}) \cdot \min\left( \frac{L(\boldsymbol{H})}{\sum_{i \in [k]} \frac{k-1}{\xi_i}}, \left( \min_{i,j \in [k]} \xi_i L_{ij}(\boldsymbol{H}) \right)^2 \right)$, Algorithm 1 which consists of $((A_t), \tau, J)$ has the expected stopping time upper bound as follows:

$$\mathbb{E}_{\boldsymbol{\mu} \sim \boldsymbol{H}}[\tau] \leq C \cdot \sigma_{\max}^2 \frac{L(\boldsymbol{H})^2}{\delta} \log\left( \frac{L(\boldsymbol{H})}{\delta} \right) + O(\log \delta^{-1}), \tag{6}$$

where $C = 320\left( \frac{\pi^2}{3} + 1 \right)$ is a universal constant and $\sigma_{\max} = \max_{i \in [k]} \sigma_i$. Here, $O(\log \delta^{-1})$ is a function of $\delta$ and $\boldsymbol{H}$ that is proportional to $\log \delta^{-1}$ when we view prior parameters $\boldsymbol{H}$ as constants. Plus, the strategy defined by Algorithm 1 is in $\mathcal{A}^b(\delta, \boldsymbol{H})$.

See Appendix E for the formal proof of Theorem 6.

**Remark 3.** When we compare the lower bound (Theorem 4) with the upper bound of Algorithm 1 (Theorem 6), we can see the algorithm is near-optimal. If we view $\sigma_{\max}/\sigma_{\min}$ as a constant, the bounds are tight up to a $\log \frac{L(\boldsymbol{H})}{\delta}$ factor.

**Remark 4.** The condition $\delta < 4L(\boldsymbol{H}) \cdot \min\left( L(\boldsymbol{H})/\sum_{i \in [k]} \frac{k-1}{\xi_i}, \min_{i,j \in [k]} (\xi_i L_{ij}(\boldsymbol{H}))^2 \right)$ is only for cleaner illustration of the regret bound in Theorem 6. The non-asymptotic result, when $\delta$ is a moderately large constant, can be found in Appendix E.1.

**Proof sketch of Theorem 6** We summarize the general strategy for the proof as follows. By the law of total expectation, $\mathbb{E}_{\boldsymbol{\mu} \sim \boldsymbol{H}}[\tau] = \mathbb{E}_{\boldsymbol{\mu} \sim \boldsymbol{H}}\left[ \mathbb{E}_{\boldsymbol{\mu}}[\tau] \right]$. Therefore, we first derive a frequentist upper bound of $\mathbb{E}_{\boldsymbol{\mu}}[\tau]$, and then marginalize it to obtain the expected Bayesian stopping time.

First, with the confidence bound defined as Eq. (5) we have the following guarantee that the true means for all arms are in the confidence bound interval with high probability.

**Lemma 7.** For any fixed $\boldsymbol{\mu} \in \{\mathbf{v} \in \mathbb{R}^k : v_i \neq v_j \text{ for all } i,j \in [k]\}$, let $\mathcal{X}(\boldsymbol{\mu}) := \{\forall i \in [k] \text{ and } t \in \mathbb{N}, \ \mu_i \in (\text{LCB}(i,t), \text{UCB}(i,t))\}$. Then, $\mathbb{P}_{\boldsymbol{\mu}}[\mathcal{X}(\boldsymbol{\mu})] \geq 1 - \delta^2$.

Now we can rewrite $\mathbb{E}_{\boldsymbol{\mu} \sim \boldsymbol{H}}[\tau]$ as follows:

$$
\begin{aligned}
\mathbb{E}_{\boldsymbol{\mu} \sim \boldsymbol{H}}[\tau] &= \mathbb{E}_{\boldsymbol{\mu} \sim \boldsymbol{H}}[\mathbb{E}_{\boldsymbol{\mu}}[\tau]] && \text{(Law of Total Expectation)} \\
&= \mathbb{E}_{\boldsymbol{\mu} \sim \boldsymbol{H}}[\mathbb{E}_{\boldsymbol{\mu}}[\tau \mathbf{1}[\mathcal{X}(\boldsymbol{\mu})]]] + \mathbb{E}_{\boldsymbol{\mu} \sim \boldsymbol{H}}[\mathbb{E}_{\boldsymbol{\mu}}[\tau \mathbf{1}[\mathcal{X}(\boldsymbol{\mu})^c]]] \\
&= \sum_i \mathbb{E}_{\boldsymbol{\mu} \sim \boldsymbol{H}}\Big[\mathbb{E}_{\boldsymbol{\mu}}[N_i(\tau)\mathbf{1}[\mathcal{X}(\boldsymbol{\mu})]]\Big] + \mathbb{E}_{\boldsymbol{\mu} \sim \boldsymbol{H}}[\mathbb{E}_{\boldsymbol{\mu}}[\tau \mathbf{1}[\mathcal{X}(\boldsymbol{\mu})^c]]]. && (7)
\end{aligned}
$$

Let $\Delta_i = \Delta_i(\boldsymbol{\mu}) := (\max_{s \in [k]} \mu_s) - \mu_i$ and $R_0(\Delta) \approx \lceil C\sigma_{\max}^2 \cdot \frac{\log \Delta^{-1}}{\Delta^2}\rceil$. For the first term, under $\mathcal{X}(\boldsymbol{\mu})$, we can bound $N_i(\tau)$ by $R_0(\max(\Delta_0, \Delta_i))$ (Lemma 17 in Appendix E), and integrate it over the prior distribution to obtain the leading factor. For the second term, thanks to the indifference stopping condition ($\hat{\Delta}^{\text{safe}}(t) \leq \Delta_0$), one can prove that $\tau$ is always smaller than $R(\Delta_0)$ (Lemma 14 in Appendix E), which leads to non-leading term.

To check that the expected probability of error is below $\delta$, we have an additional lemma:

**Lemma 8** (Probability of dropping $i^*(\boldsymbol{\mu})$). For any $\boldsymbol{\mu}_0$, under $\mathcal{H}_{\boldsymbol{\mu}_0}$, $\mathcal{X}(\boldsymbol{\mu}_0) \subset \bigcap_t \{i^*(\boldsymbol{\mu}_0) \in \mathcal{A}(t)\}$.

This lemma means under the event $\mathcal{X}(\boldsymbol{\mu})$, the best arm is never dropped. We can also prove that under the event $\mathcal{X}(\boldsymbol{\mu})$, if $\Delta_i(\boldsymbol{\mu}) > \Delta_0$, the sub-optimal arm will eventually be dropped before the algorithm terminates (Lemma 14 in Appendix E). These two facts mean there are only two cases in which the prediction of Algorithm 1 could be wrong.

- Under $\mathcal{X}(\boldsymbol{\mu})^c$, both facts cannot guarantee the correct identification. From Lemma 7, $\mathbb{P}_{\boldsymbol{\mu}}[\mathcal{X}(\boldsymbol{\mu})^c] \leq \delta^2$ for all $\boldsymbol{\mu}$, and thus $\mathbb{P}_{\boldsymbol{\mu} \sim \boldsymbol{H}}[\mathcal{X}(\boldsymbol{\mu})^c] \leq \delta^2$.
- When $\Delta_i(\boldsymbol{\mu}) \leq \Delta_0$. From Lemma 1 and the definition of $\Delta_0$, the probability of drawing such $\boldsymbol{\mu}$ from the prior is at most $\delta/2$.

Therefore, by union bound, Algorithm 1 has the expected probability of misidentification guarantee smaller than $\delta^2 + \delta/2 < \delta$.

# 6 Simulation

We conduct two experiments to demonstrate that the expected stopping times of frequentist $\delta$-correct algorithms diverge in a Bayesian setting and that the elimination process in Algorithm 1 is necessary for more efficient sampling. In Tables 1 and 2, each column 'Avg', 'Max', and 'Error' represents the average stopping time, maximum stopping time, and the ratio of the misidentification, respectively.[5] More details of these experiments are in Appendix F.

**Frequentist algorithms diverge in Bayesian Setting**  We evaluate the empirical performance of our Elimination algorithm (Algorithm 1) by comparing it with other frequentist algorithms such as Top-two Thompson Sampling (TTTS) [Russo, 2016] and Top-two UCB (TTUCB) [Jourdan and Degenne, 2022b].

We design an experiment setup that has $k = 2$ arms with standard Gaussian prior distribution, which means $m_i = 0, \xi_i = 1$ for all $i \in [k]$. We set $\delta = 0.1$ and ran $N = 1000$ Bayesian FC-BAI simulations to estimate the expected stopping time and success rate.

In Table 1, one can see that the two top-two algorithms exhibit very large maximum stopping time. This supports our theoretical result in Section 3 that the expected stopping time of Frequentist $\delta$-correct algorithms will diverge in the Bayesian setting. We did not check the track and stop algorithm [Garivier and Kaufmann, 2016] because it needs to solve an optimization for each round, but the fact that the expected stopping time of the track and stop is at least half of the TTTS and TTUCB for a small $\delta$ implies that the performance of track and stop is similar to that of top-two algorithms. Algorithm 1 shows a significantly smaller average stopping time as well as an average computation time than that of these algorithms.

---

[5]We include the computation time in the Appendix F.3

Table 1: Comparison of two top-two algorithms and Algorithm 1.

|         | AVG               | MAX               | ERROR |
|---------|-------------------|-------------------|-------|
| ALG. 1  | $1.06 \times 10^4$ | $2.35 \times 10^5$ | 1.5%  |
| TTTS    | $1.56 \times 10^5$ | $1.09 \times 10^8$ | 0.5%  |
| TTUCB   | $1.95 \times 10^5$ | $1.13 \times 10^8$ | 0%    |

Table 2: Comparison of Algorithm 1 and the no-elimination version of it.

|         | AVG               | MAX               | ERROR |
|---------|-------------------|-------------------|-------|
| ALG. 1  | $2.69 \times 10^5$ | $1.66 \times 10^7$ | 0.6%  |
| NOELIM  | $1.29 \times 10^6$ | $8.25 \times 10^7$ | 0%    |

**Effect of the elimination process**   We implemented the modification of Algorithm 1 (denoted as NoElim) that never eliminates an arm from $\mathcal{A}(t)$ [6] In this setup, we have $k = 10$ arms with standard Gaussian prior distribution, which means $m_i = 0, \xi_i = 1$ for all $i \in [k]$. We set $\delta = 0.01$ and ran $N = 1000$ Bayesian FC-BAI simulations.

As one can check from Table 2, elimination of arms helps the efficient use of samples and reduces stopping time and computation time.

## 7   Discussion and future works

We have considered the Gaussian Bayesian best arm identification with fixed confidence. We show that the traditional Frequentist FC-BAI algorithms do not stop in finite time in expectation, which implies the suboptimality of such algorithms in the Bayesian FC-BAI problem. We have established a lower bound of the Bayesian expected stopping time, which is of order $\Omega(\frac{L(\boldsymbol{H})^2}{\delta})$. Moreover, we have introduced the elimination and early stopping algorithm, which achieves a matching stopping time up to a polylogarithmic factor of $L(\boldsymbol{H})$ and $\delta$. We conduct simulations to support our results.

In the future, we will attempt to tighten the logarithmic and $\left(\frac{\max_i \sigma_i}{\min_i \sigma_i}\right)^2$ gap between the lower and upper bound, extend the indifference zone strategy for other traditional BAI algorithms in the Bayesian setting, extend our analysis from Gaussian bandit instances to general exponential families, and design a robust algorithm against misspecified priors.

## Acknowledgements

K. Jang acknowledge the financial support from the MUR PRIN grant 2022EKNE5K (Learning in Markets and Society), the FAIR (Future Artificial Intelligence Research) project, funded by the NextGenerationEU program within the PNRR-PE-AI scheme, and the the EU Horizon CL4-2022-HUMAN-02 research and innovation action under grant agreement 101120237, project ELIAS (European Lighthouse of AI for Sustainability).

J. Komiyama was supported by NYU Stern School of Business Research Scholars Fund no. 10-83004-BF478.

K. Yamazaki was supported by JSPS KAKENHI grant no. JP20K03758, JP24K06844 and JP24H00328 and the start-up grant by the School of Mathematics and Physics of the University of Queensland.

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

# A  Notation table

Table 3: Major notation

| symbol | definition |
|---|---|
| $k$ | number of the arms |
| $\delta$ | confidence level |
| $\boldsymbol{\mu}$ | means $(= (\mu_1, \mu_2, \ldots, \mu_k))$ |
| $\boldsymbol{H}$ | prior distribution of $\boldsymbol{\mu}$ |
| $H_i$ | prior distribution of $\mu_i$ |
| $h_i$ | prior density of $\mu_i$ |
| $m_i, \xi_i$ | mean and standard deviation of $h_i$ |
| $N_i(t)$ | $\sum_{s=1}^{t-1} \mathbf{1}[A_s = i]$ |
| $L(\boldsymbol{H})$ | See Definition 2 |
| $i^*(\boldsymbol{\mu}), j^*(\boldsymbol{\mu})$ | best arm and second best arm |
| $\boldsymbol{\nu}(\boldsymbol{\mu})$ | alternative model where the top-two means of $\boldsymbol{\mu}$ are swapped |
| $\mathrm{KL}_i(\cdot, \cdot)$ | KL divergence between two distributions |
| $d(p, q)$ | KL divergence between two Bernoulli distributions with parameters $p$ and $q$ |
| $N_i(t)$ | number of draws on arm $i$ before time step $t$ |
| $B$ | $320 \max_{i \in [k]} \sigma_i^2$ |
| $B_0$ | $\left( \frac{\pi^2}{3} + 1 \right) B$ |
| $\Theta_i$ | $\{\boldsymbol{\mu} \in \mathbb{R}^k : i^*(\boldsymbol{\mu}) = i\}$ |
| $\Theta_{ij}$ | $\{\boldsymbol{\mu} \in \mathbb{R}^k : i^*(\boldsymbol{\mu}) = i, j^*(\boldsymbol{\mu}) = j\}$ |
| $\boldsymbol{\mu}_{\backslash i}$ ( $\boldsymbol{\mu}_{\backslash i,j}$) | the vector projection which omits $i$-th coordinate ($i, j$-th, respectively) |
| $\boldsymbol{H}_{\backslash i}$ ( $\boldsymbol{H}_{\backslash i,j}$) | the distribution which omits $i$-th coordinate ($i, j$-th, respectively) |

Table 4: Notations for the lower bound proof, Section D

| symbol | definition |
|---|---|
| $R$ | $e^{-4}$ |
| $\tilde{\Delta}$ | $\frac{32 e^4}{L(\boldsymbol{H})} \delta$ |
| $\boldsymbol{\nu}(\boldsymbol{\mu})$ | alternative model where the top-two means are swapped |
| $n_i(\boldsymbol{\mu})$ | $\mathbb{E}_{\boldsymbol{\mu}}[N_i(\tau)]$ |
| $D_0(\boldsymbol{H})$ | $\begin{cases} W(-\frac{1}{32 \max_{i \in [k]} \xi_i^{3/2}}) & \text{If } \max_{i \in [k]} \xi_i > \sqrt[3]{\frac{e^2}{2^{10}}} \\ 1 & \text{Otherwise} \end{cases}$ ($W$ is the Lambert W function.) |
| $D_1(\boldsymbol{H})$ | $\min_{i \neq j} \left[ \left\| \frac{m_i}{\sigma_i^2} - \frac{m_j}{\sigma_j^2} \right\|^{-1}, \left[ \frac{1}{2\sigma_i^2} + \frac{1}{2\sigma_j^2} \right]^{-2} \right]$ |
| $\delta_L(\boldsymbol{H})$ | $\frac{L(\boldsymbol{H})}{32 e^4} \cdot \min \left( D_0(\boldsymbol{H}), D_1(\boldsymbol{H}), \min_{i \in [k]} \frac{1}{4 m_i^2}, \frac{L(\boldsymbol{H})}{4(k-1) \sum_{i \in [k]} \frac{1}{\xi_i}} \right)$ |

# B  Proof of Lemma 1

We will use the following formal version of the volume lemma for the proof. The first result is used for the upper bound, and the second result is used for the lower bound.

**Lemma 9** (Volume Lemma, formal). *Let $\Theta_{ij} := \{\boldsymbol{\mu} \in \mathbb{R}^k : i^*(\boldsymbol{\mu}) = i, j^*(\boldsymbol{\mu}) = j\}$.*

    1. *For any $\Delta \in (0, 1)$, define*

$$L_{ij}(\boldsymbol{H}, \Delta) := \frac{1}{\delta} \int_{\Theta_{ij}} \mathbf{1}[|\mu_i - \mu_j| \leq \Delta] \, d\boldsymbol{H}(\boldsymbol{\mu}).$$

Table 5: Notations for the upper bound proof, Section E

| symbol | definition |
|---|---|
| $\mathrm{Conf}(i,t)$ | confidence bound (Section 5) |
| $\mathrm{UCB}(i,t), \mathrm{LCB}(i,t)$ | upper and lower confidence bounds (Section 5) |
| $\hat{\Delta}^{\mathrm{safe}}(t)$ | $\max_{i\in\mathcal{A}(t)} \mathrm{UCB}(i,t) - \max_{i\in\mathcal{A}(t)} \mathrm{LCB}(i,t)$ |
| $\Delta_0$ | $\frac{\delta}{4L(\boldsymbol{H})}$ |
| $\Delta_{thr}$ | $\min\left(\log\frac{4\sqrt{k}}{\delta\pi}, \frac{1}{B}\right)$ |
| $R_0(\Delta)$ | $B\frac{\log\min(\Delta,\Delta_{thr})^{-1}}{\min(\Delta,\Delta_{thr})^2}$ |
| $T_0$ | $kR_0(\Delta_0)$ |
| $\Delta_s(\boldsymbol{\mu})$ | $\mu_{i^*(\boldsymbol{\mu})} - \mu_s$ |
| $\mathcal{X}(\boldsymbol{\mu})$ | Event $\bigcap_{i\in[k]}\left[\left(\bigcap_{t=1}^{\infty}\{\mathrm{LCB}(i,t)\le\mu_i\}\right)\bigcap\left(\bigcap_{t=1}^{\infty}\{\mathrm{UCB}(i,t)\ge\mu_i\}\right)\right].$ (Eq. (24)) |

Then,

$$L_{ij}(\boldsymbol{H}, \Delta) \in \left[L_{ij}(\boldsymbol{H}) - \frac{1}{\xi_i}\Delta, L_{ij}(\boldsymbol{H}) + \frac{1}{\xi_i}\Delta\right]. \tag{8}$$

In particular, when $\Delta < \frac{L(\boldsymbol{H})}{\sum_{i\in[k]}\frac{k-1}{\xi_i}}$, $L(\boldsymbol{H}, \Delta) \le 2L(\boldsymbol{H})$.

2. (Volume lemma for the lower bound) For small enough positive real number $\Delta < \min\left[\frac{1}{4\max_{i\in[k]}m_i^2}, D_0(\boldsymbol{H})\right]$[7] let

$$L'_{ij}(\boldsymbol{H}, \Delta) := \frac{1}{\Delta}\int_{\Theta_{ij}} \mathbf{1}[|\mu_i - \mu_j| \le \Delta]\mathbf{1}[|\mu_i|, |\mu_j| \le \frac{1}{\sqrt{\Delta}}]\,\mathrm{d}\boldsymbol{H}(\boldsymbol{\mu}).$$

Then,

$$L'_{ij}(\boldsymbol{H}, \Delta) \in \left[L_{ij}(\boldsymbol{H}) - \frac{2}{\xi_i}\Delta, L_{ij}(\boldsymbol{H}) + \frac{1}{\xi_i}\Delta\right]. \tag{9}$$

In particular, when $\Delta < \min\left[\frac{1}{4\max_{i\in[k]}m_i^2}, D_0(\boldsymbol{H}), \frac{L(\boldsymbol{H})}{\sum_{i\in[k]}\frac{4(k-1)}{\xi_i}}\right]$, $L'(\boldsymbol{H}, \Delta) \in \left[\frac{1}{2}L(\boldsymbol{H}), 2L(\boldsymbol{H})\right]$.

*Proof.* First, let us prove the upper bound of Eq. (8), i.e. $L_{ij}(\boldsymbol{H}, \delta) \le L_{ij}(\boldsymbol{H}) + \frac{1}{\xi_i}\Delta$.

Recall for $i, j \in [k]$, $\Theta_{ij} = \{\mu : i^*(\mu) = i, j^*(\mu) = j\}$. We have

$$\int_{\Theta_{ij}} \mathbf{1}\left[|\mu_i - \mu_j| \le \Delta\right]\mathrm{d}\boldsymbol{H}(\boldsymbol{\mu}) = \int_{-\infty}^{\infty}\int_{\mu_j}^{\mu_j+\Delta} h_i(\mu_i)\,\mathrm{d}\mu_i \prod_{k\ne i,j}\int_{-\infty}^{\mu_j} h_k(\mu_k)\,\mathrm{d}\mu_k h_j(\mu_j)\,\mathrm{d}\mu_j$$

$$\le \int_{-\infty}^{\infty}\left(\Delta\max_{0\le y\le\Delta} h_i(\mu_j+y)\right)\prod_{k\ne i,j}\int_{-\infty}^{\mu_j} h_k(\mu_k)\,\mathrm{d}\mu_k h_j(\mu_j)\,\mathrm{d}\mu_j$$

$$\le \Delta\int_{-\infty}^{\infty}\left[h_i(\mu_j) + \frac{e^{-1/2}}{\xi_i}\Delta\right]\prod_{k\ne i,j}\int_{-\infty}^{\mu_j} h_k(\mu_k)\,\mathrm{d}\mu_k h_j(\mu_j)\,\mathrm{d}\mu_j$$

(by the Lipschitz property of the Gaussian density, $e^{-1/2}/\xi_i$ is the steepest slope of $N(m_i, \xi_i^2)$)

$$\le \Delta\left(\underbrace{\int_{-\infty}^{\infty} h_i(\mu_j)\prod_{k\ne i,j}\int_{-\infty}^{\mu_j} h_k(\mu_k)\,\mathrm{d}\mu_k h_j(\mu_j)\,\mathrm{d}\mu_j}_{=L_{ij}(\boldsymbol{H})} + \left[\frac{e^{-1/2}}{\xi_i}\Delta\right]\right).$$

---

[7]See Appendix G for the definition of $D_0(\boldsymbol{H})$.

Therefore, we verified the upper bound side of Eq. (8).

For the lower bound, by following the same steps, one can prove $\int_{\Theta_{ij}} \mathbf{1}\left[|\mu_i - \mu_j| \le \Delta\right] \mathrm{d}\boldsymbol{H}(\boldsymbol{\mu}) \ge (L_{ij}(\boldsymbol{H}) - \frac{1}{\xi_i}\Delta)\Delta$.

For the proof of Eq. (9), by Chernoff's method we can bound the tail probability as follows:

$$\int_{\Theta_{ij}} \mathbf{1}\left[|\mu_i - m_i| > \frac{1}{\sqrt{\Delta}}\right]\mathrm{d}H(\boldsymbol{\mu}) < \int_{\mathbb{R}^k} \mathbf{1}\left[|\mu_i - m_i| > \frac{1}{\sqrt{\Delta}}\right]\mathrm{d}H(\boldsymbol{\mu}) < 2\exp\left(-\frac{1}{2\Delta\xi_i^2}\right).$$

When $|m_i| < \frac{1}{2\sqrt{\Delta}}$, then we can change the above inequality as:

$$\int_{\Theta_{ij}} \mathbf{1}\left[|\mu_i| > \frac{1}{\sqrt{\Delta}}\right]\mathrm{d}H(\boldsymbol{\mu}) \le \int_{\mathbb{R}^k} \mathbf{1}\left[|\mu_i| > \frac{1}{\sqrt{\Delta}}\right]\mathrm{d}H(\boldsymbol{\mu})$$

$$\le \int_{\mathbb{R}^k} \mathbf{1}\left[|\mu_i - m_i| > \frac{1}{2\sqrt{\Delta}}\right]\mathrm{d}H(\boldsymbol{\mu}) < 2\exp\left(-\frac{1}{8\Delta\xi_i^2}\right).$$

Therefore,

$$\int_{\Theta_{ij}} \mathbf{1}[|\mu_i - \mu_j| \le \Delta]\mathbf{1}[|\mu_i|, |\mu_j| \le \frac{1}{\sqrt{\Delta}}]\mathrm{d}\boldsymbol{H}(\boldsymbol{\mu}) \ge \int_{\Theta_{ij}} \mathbf{1}\left[|\mu_i - \mu_j| \le \Delta\right]\mathrm{d}\boldsymbol{H}(\boldsymbol{\mu})$$

$$- \int_{\Theta_{ij}} \mathbf{1}\left[|\mu_i - m_i| > \frac{1}{2\sqrt{\Delta}}\right]\mathrm{d}H(\boldsymbol{\mu}) - \int_{\Theta_{ij}} \mathbf{1}\left[|\mu_j - m_j| > \frac{1}{2\sqrt{\Delta}}\right]\mathrm{d}H(\boldsymbol{\mu})$$

$$\ge (L_i(\boldsymbol{H}) - \frac{1}{\xi_i}\Delta)\Delta - 2\exp\left(-\frac{1}{8\Delta\xi_i^2}\right) - 2\exp\left(-\frac{1}{8\Delta\xi_j^2}\right)$$

$$\ge (L_i(\boldsymbol{H}) - \frac{2}{\xi_i}\Delta)\Delta \qquad\qquad \text{(by } \Delta < D_0(\boldsymbol{H}))$$

for $\Delta < \min\left[\min_{i \in [k]} \frac{1}{4m_i^2}, D_0(\boldsymbol{H})\right]$. □

## C   Proof of Theorem 2

For $\Delta > 0$, let $\Theta_i(\Delta) := \{\boldsymbol{\mu} \in \mathbb{R}^k : i^*(\boldsymbol{\mu}) = i, \mu_i - \mu_{j^*(\boldsymbol{\mu})} \le \Delta\}$. From Lemma 9, we have

$$\mathbb{P}_{\boldsymbol{\mu}\sim\boldsymbol{H}}\left[i^*(\boldsymbol{\mu}) = i, \mu_i - \mu_{j^*(\boldsymbol{\mu})} \le \Delta\right] = \sum_{j \ne i} \mathbb{P}_{\boldsymbol{\mu}\sim\boldsymbol{H}}\left[i^*(\boldsymbol{\mu}) = i, j^*(\boldsymbol{\mu}) = j, \mu_i - \mu_j \le \Delta\right]$$

$$\ge \sum_{j \ne i} \frac{1}{2}L_{ij}(\boldsymbol{H})\Delta.$$

Now, for each $\boldsymbol{\mu} \in \mathbb{R}^k$, let $\nu : \mathbb{R}^k \to \mathbb{R}^k$ be a function such that for $s \in [k]$,

$$\nu(\boldsymbol{\mu})_s := \begin{cases} \mu_{i^*(\boldsymbol{\mu})} & \text{when } s = j^*(\boldsymbol{\mu}) \\ \mu_{j^*(\boldsymbol{\mu})} & \text{when } s = i^*(\boldsymbol{\mu}) \\ \mu_s & \text{Otherwise.} \end{cases}$$

For any $\boldsymbol{\mu} \in \mathbb{R}^k$, let $\mathcal{E}(\boldsymbol{\mu}) = \{J \ne i^*(\boldsymbol{\mu})\}$ and $\boldsymbol{\nu} = \nu(\boldsymbol{\mu})$. By Lemma 1 in Kaufmann et al. [2016a], for any frequentist $\delta$-correct algorithm $\pi = ((A_t)_t, \tau, J)$, we have

$$\sum_{s \in [k]} \mathbb{E}_{\boldsymbol{\mu}}[N_s(\tau)]\mathrm{KL}_s(\mu_s || \nu_s) \ge d(P_{\boldsymbol{\mu}}(\mathcal{E}(\boldsymbol{\mu})), P_{\boldsymbol{\nu}}(\mathcal{E}(\boldsymbol{\mu}))), \quad \boldsymbol{\mu} \in \mathbb{R}^k. \tag{10}$$

For the left side, from the construction of $\boldsymbol{\nu}$,

$$\mathrm{KL}_s(\mu_s||\nu_s) := \begin{cases} \frac{(\mu_{i^*(\boldsymbol{\mu})} - \mu_{j^*(\boldsymbol{\mu})})^2}{2\sigma_s^2} & s = i^*(\boldsymbol{\mu}), j^*(\boldsymbol{\mu}) \\ 0 & \text{Otherwise.} \end{cases}$$

Since $\pi \in \mathcal{A}^f(\delta)$ and from the definition of $\mathcal{E}(\boldsymbol{\mu})$ and $\boldsymbol{\nu}$, $\mathbb{P}_{\boldsymbol{\mu}}(\mathcal{E}(\boldsymbol{\mu})) \le \delta$ and $\mathbb{P}_{\boldsymbol{\nu}}(\mathcal{E}(\boldsymbol{\mu})) \ge 1 - \delta$. Overall, we can rewrite Eq. (10) to the following simpler form:

$$\mathbb{E}_{\boldsymbol{\mu}}[N_{i^*(\boldsymbol{\mu})}(\tau)] \frac{(\mu_{i^*(\boldsymbol{\mu})} - \mu_{j^*(\boldsymbol{\mu})})^2}{2\sigma_{i^*(\boldsymbol{\mu})}^2} + \mathbb{E}_{\boldsymbol{\mu}}[N_{j^*(\boldsymbol{\mu})}(\tau)] \frac{(\mu_{i^*(\boldsymbol{\mu})} - \mu_{j^*(\boldsymbol{\mu})})^2}{2\sigma_{j^*(\boldsymbol{\mu})}^2} \ge d(\delta, 1 - \delta)$$

$$\implies \mathbb{E}_{\boldsymbol{\mu}}[N_{i^*(\boldsymbol{\mu})}(\tau) + N_{j^*(\boldsymbol{\mu})}(\tau)] \ge \frac{2d(\delta, 1 - \delta) \min_{s \in [k]} \sigma_s^2}{(\mu_{i^*(\boldsymbol{\mu})} - \mu_{j^*(\boldsymbol{\mu})})^2}.$$

Since $\tau = \sum_{s=1}^k N_s(\tau)$, we can lower bound the expected stopping time when $\boldsymbol{\mu}$ is given as follows:

$$\mathbb{E}_{\boldsymbol{\mu}}[\tau] \ge \mathbb{E}_{\boldsymbol{\mu}}[N_{i^*(\boldsymbol{\mu})}(\tau) + N_{j^*(\boldsymbol{\mu})}(\tau)] \ge \frac{2d(\delta, 1 - \delta) \min_{s \in [k]} \sigma_s^2}{(\mu_{i^*(\boldsymbol{\mu})} - \mu_{j^*(\boldsymbol{\mu})})^2}. \tag{11}$$

Now, when we compute marginal $\mathbb{E}_{\boldsymbol{\mu}}[\tau]$ over $\boldsymbol{\mu}$, we have

$$\begin{aligned} \mathbb{E}_{\boldsymbol{\mu} \sim \boldsymbol{H}}[\tau] &= \mathbb{E}_{\boldsymbol{\mu} \sim \boldsymbol{H}}\left[\mathbb{E}_{\boldsymbol{\mu}}[\tau]\right] && \text{(Law of total expectation)} \\ &\ge \mathbb{E}_{\boldsymbol{\mu} \sim \boldsymbol{H}}\left[\mathbb{E}_{\boldsymbol{\mu}}[\tau]\mathbf{1}_{\Theta_i(\Delta)}\right] \\ &\ge \mathbb{E}_{\boldsymbol{\mu} \sim \boldsymbol{H}}\left[\frac{2d(\delta, 1 - \delta) \min_{s \in [k]} \sigma_s^2}{(\mu_{i^*(\boldsymbol{\mu})} - \mu_{j^*(\boldsymbol{\mu})})^2}\mathbf{1}_{\Theta_i(\Delta)}\right] && \text{(Eq. (11))} \\ &\ge \mathbb{E}_{\boldsymbol{\mu} \sim \boldsymbol{H}}\left[\frac{2d(\delta, 1 - \delta) \min_{s \in [k]} \sigma_s^2}{\Delta^2}\mathbf{1}_{\Theta_i(\Delta)}\right] \\ &\ge \frac{\left(\sum_{j \ne i} L_{ij}(\boldsymbol{H})\right) d(\delta, 1 - \delta) \min_{s \in [k]} \sigma_s^2}{\Delta^2} = \frac{\left(\sum_{j \ne i} L_{ij}(\boldsymbol{H})\right) d(\delta, 1 - \delta) \min_{s \in [k]} \sigma_s^2}{\Delta}. \end{aligned}$$

Now since $\Delta$ is an arbitrary small positive number, we can conclude that $\mathbb{E}_{\boldsymbol{\mu} \sim \boldsymbol{H}}[\tau]$ diverges.

## D  Proof of Theorem 4

In this subsection, we will prove the following theorem:

**Theorem 10** (Restatement of Theorem 4). Let $\delta > 0$ be sufficiently small such that $\delta < \delta_L(\boldsymbol{H})$. For any best arm identification algorithm, if

$$\int_{\mathbb{R}^k} \left(\sum_{i \in [k]} n_i(\boldsymbol{\mu})\right) \mathrm{d}\boldsymbol{H}(\boldsymbol{\mu}) \le N_V,$$

then

$$\int_{\mathbb{R}^k} \mathbb{P}_{\boldsymbol{\mu}}[J \ne i^*(\boldsymbol{\mu})] \mathrm{d}\boldsymbol{H}(\boldsymbol{\mu}) \ge \delta.$$

*Proof.* By Lemma 1 in Kaufmann et al. [2016a], for any stopping time $\tau$, we have

$$\sum_{i \in [k]} \mathbb{E}_{\boldsymbol{\mu}}[N_i(\tau)]\mathrm{KL}_i(\mu_i||\nu_i) \ge d(\mathbb{P}_{\boldsymbol{\mu}}(\mathcal{E}(\boldsymbol{\mu})), \mathbb{P}_{\boldsymbol{\nu}}(\mathcal{E}(\boldsymbol{\mu}))), \quad \boldsymbol{\mu} \in \mathbb{R}^k. \tag{12}$$

To modify the RHS of Eq. (12), we will use the following lemma:

**Lemma 11.** For any $p, q', q \in (0, 1)$ such that $q' \leq q$, we have

$$\ln \frac{1}{4(p+q)} \leq d(p, 1-q').$$

By using Lemma 11 and the fact that $\mathbb{P}_{\boldsymbol{\nu}}(\mathcal{E}(\boldsymbol{\mu})) \geq 1 - \mathbb{P}_{\boldsymbol{\nu}}[J \neq i^*(\boldsymbol{\nu})]$ by definition, we can transform (12) into

$$\sum_{i \in [k]} \mathbb{E}_{\boldsymbol{\mu}}[N_i(\tau)] \mathrm{KL}_i(\mu_i || \nu_i) \geq \ln \frac{1}{4(\mathbb{P}_{\boldsymbol{\mu}}[J \neq i^*(\boldsymbol{\mu})] + \mathbb{P}_{\boldsymbol{\nu}}[J \neq i^*(\boldsymbol{\nu})])}, \quad \forall \boldsymbol{\mu} \in \mathbb{R}^k. \qquad (13)$$

Note that $\mathbb{P}_{\boldsymbol{\mu}}(\mathcal{E}(\boldsymbol{\mu}))$ is exactly the error probability, and we are interested in the marginal error probability $\mathbb{E}_{\boldsymbol{\mu} \sim H}[\mathbb{P}_{\boldsymbol{\mu}}(\mathcal{E})]$. Let $n_i(\boldsymbol{\mu}) = \mathbb{E}_{\boldsymbol{\mu}}[N_i(\tau)]$, and $\tilde{\Delta}$ is an arbitrary small enough positive variable which we will define later on Eq. (19). Then, by rearrangement, we can induce the following inequalities

$$\int_{\mathbb{R}^k} \mathbb{P}_{\boldsymbol{\mu}}[J \neq i^*(\boldsymbol{\mu})] \, \mathrm{d}\boldsymbol{H}(\boldsymbol{\mu})$$

$$= \sum_{i \in [k]} \sum_{j \neq i} \int_{\Theta_{ij}} \mathbb{P}_{\boldsymbol{\mu}}[J \neq i] \, \mathrm{d}\boldsymbol{H}(\boldsymbol{\mu})$$

$$= \frac{1}{2} \left( \sum_{i \in [k]} \sum_{j \neq i} \int_{\Theta_{ij}} \mathbb{P}_{\boldsymbol{\mu}}[J \neq i] \, \mathrm{d}\boldsymbol{H}(\boldsymbol{\mu}) + \sum_{j \in [k]} \sum_{i \neq j} \int_{\Theta_{ji}} \mathbb{P}_{\boldsymbol{\nu}(\boldsymbol{\mu})}[J \neq j] h_i(\mu_j) h_j(\mu_i) \left( \prod_{s \neq i,j} h_s(\mu_s) \right) \mathrm{d}\boldsymbol{\mu} \right)$$

(Symmetry of the Lebesgue measure)

$$= \frac{1}{2} \left( \sum_{i \in [k]} \sum_{j \neq i} \int_{\Theta_{ij}} \left[ \mathbb{P}_{\boldsymbol{\mu}}[J \neq i] h_i(\mu_i) h_j(\mu_j) + \mathbb{P}_{\boldsymbol{\nu}(\boldsymbol{\mu})}[J \neq j] h_i(\mu_j) h_j(\mu_i) \right] \left( \prod_{s \neq i,j} h_s(\mu_s) \right) \mathrm{d}\boldsymbol{\mu} \right)$$

$$\geq \sum_{i \in [k]} \sum_{j \neq i} \int_{\Theta_{ij}} \frac{\mathbb{P}_{\boldsymbol{\mu}}[J \neq i^*(\boldsymbol{\mu})] + \mathbb{P}_{\boldsymbol{\nu}}[J \neq i^*(\boldsymbol{\nu})]}{2} \min \left( h_i(\mu_i) h_j(\mu_j), h_i(\mu_j) h_j(\mu_i) \right) \mathrm{d}\boldsymbol{\mu}$$

$$(AC + BD \geq A \min(C, D) + B \min(C, D) = (A + B) \min(C, D) \text{ for } A, B, C, D > 0)$$

$$\geq \sum_{i \in [k]} \sum_{j \neq i} \int_{\Theta_{ij}} \frac{\exp\left(-n_i(\boldsymbol{\mu}) \mathrm{KL}_i(\mu_i, \mu_j) - n_j(\boldsymbol{\mu}) \mathrm{KL}_j(\mu_j, \mu_i)\right)}{8} \min \left( 1, \frac{h_i(\mu_j) h_j(\mu_i)}{h_i(\mu_i) h_j(\mu_j)} \right) h_i(\mu_i) h_j(\mu_j) \, \mathrm{d}\boldsymbol{\mu}$$

(Eq. (13))

$$\geq e^{-4} \sum_{i \in [k]} \sum_{j \neq i} \int_{\Theta_{ij}} \frac{\exp\left(-n_i(\boldsymbol{\mu}) \mathrm{KL}_i(\mu_i, \mu_j) - n_j(\boldsymbol{\mu}) \mathrm{KL}_j(\mu_j, \mu_i)\right)}{8} \mathbf{1} \left[ |\mu_i|, |\mu_j| \leq \frac{1}{\sqrt{\tilde{\Delta}}} \right] \mathrm{d}H(\boldsymbol{\mu}).$$

(Lemma 12)

For the last inequality, we used the following lemma. The proof for this lemma is found in Subsection D.1.2.

**Lemma 12** (Ratio Lemma). For all $a, b \in \mathbb{R}$ which satisfy $|a - b| \leq D_1(\boldsymbol{H})$ and $|a|, |b| \leq \frac{1}{\sqrt{D_1(\boldsymbol{H})}}$ for some fixed $D_1(\boldsymbol{H})$[8], $\frac{h_i(a) h_j(b)}{h_i(b) h_j(a)} \geq e^{-4}$ for all $i, j \in [k]$.

In short, we have

$$\int_{\mathbb{R}^k} \mathbb{P}_{\boldsymbol{\mu}}[J \neq i^*(\boldsymbol{\mu})] \, \mathrm{d}\boldsymbol{H}(\boldsymbol{\mu})$$

$$\geq e^{-4} \sum_{i \in [k]} \sum_{j \neq i} \int_{\Theta_{ij}} \frac{\exp\left(-n_i(\boldsymbol{\mu}) \mathrm{KL}_i(\mu_i, \mu_j) - n_j(\boldsymbol{\mu}) \mathrm{KL}_j(\mu_j, \mu_i)\right)}{8} \mathbf{1} \left[ |\mu_i|, |\mu_j| \leq \frac{1}{\sqrt{\tilde{\Delta}}} \right] \mathrm{d}H(\boldsymbol{\mu})$$

(14)

---

[8] $D_1(\boldsymbol{H}) := \min_{i \neq j} \left[ \left| \frac{m_i}{\sigma_i^2} - \frac{m_j}{\sigma_j^2} \right|^{-1}, \left[ \frac{1}{2\sigma_i^2} + \frac{1}{2\sigma_j^2} \right]^{-2} \right]$

and the following statement is a stronger statement than Theorem 10.

$$\text{If } \int \left( \sum_{i \in [k]} n_i(\boldsymbol{\mu}) \right) \mathrm{d}\boldsymbol{H}(\boldsymbol{\mu}) \le N_V, \text{ then (RHS of Eq. (14))} \ge \delta.$$

RHS of Eq. (14) is represented in terms of $n := (n_1, \cdots, n_k) : \mathbb{R}^k \to [0, \infty)^k$ (the expected number of arm pulls) and does hold for any algorithm given $n$. To prove the above statement, since 'set of all expected number of arm pulls' is a subset of $\{\tilde{n} : \mathbb{R}^k \to [0, \infty)\}$, it suffices to show that the optimal value $V$ of the following objective

$$V^{\min} := \inf_{\tilde{n}: \mathbb{R}^k \to [0,\infty)^k} V(\tilde{n}) \tag{15}$$

$$\text{s.t. } \int_{\mathbb{R}^k} \left( \sum_{s=1}^{k} \tilde{n}_s(\boldsymbol{\mu}) \right) \mathrm{d}\boldsymbol{H}(\boldsymbol{\mu}) \le N_V$$

$$\text{where } V(\tilde{n}) := e^{-4} \sum_{i \in [k]} \sum_{j \neq i} \int_{\Theta_{ij}} \frac{\exp\left(-\tilde{n}_i(\boldsymbol{\mu})\mathrm{KL}_i(\mu_i, \mu_j) - \tilde{n}_j(\boldsymbol{\mu})\mathrm{KL}_j(\mu_j, \mu_i)\right)}{8} \mathbf{1}\left[ |\mu_i|, |\mu_j| \le \frac{1}{\sqrt{\tilde{\Delta}}} \right] \mathrm{d}H(\boldsymbol{\mu})$$

is greater than $\delta$.

Let $\tilde{\Theta}_{ij} := \{\boldsymbol{\mu} \in \Theta_{ij} : |\mu_i - \mu_j| \le \tilde{\Delta}, |\mu_i| \le \frac{1}{\sqrt{\tilde{\Delta}}}, |\mu_j| \le \frac{1}{\sqrt{\tilde{\Delta}}}\}$. Then, it holds that $V^{\min} \ge V_1^{\min}$ where

$$V_1^{\min} := \inf_{\tilde{n}: \mathbb{R}^k \to [0,\infty)^k} V_1(\tilde{n}) \tag{16}$$

$$\text{s.t. } \sum_{i \in [k]} \sum_{j \neq i} \int_{\tilde{\Theta}_{ij}} \left( \sum_{s=1}^{k} \tilde{n}_s(\boldsymbol{\mu}) \right) \mathrm{d}\boldsymbol{H}(\boldsymbol{\mu}) \le N_V$$

$$\text{where } V_1(\tilde{n}) := e^{-4} \sum_{i \in [k]} \sum_{j \neq i} \int_{\tilde{\Theta}_{ij}} \frac{\exp\left(-\tilde{n}_i(\boldsymbol{\mu})\mathrm{KL}_i(\mu_i, \mu_j) - \tilde{n}_j(\boldsymbol{\mu})\mathrm{KL}_j(\mu_j, \mu_i)\right)}{8} \mathrm{d}H(\boldsymbol{\mu}).$$

To see this, suppose $\hat{n}$ is an optimal solution to (15). Then, by the constraint of optimization problem (15), $\int_{\mathbb{R}^k} \left( \sum_{s \in [k]} \hat{n}_s(\boldsymbol{\mu}) \right) \mathrm{d}\boldsymbol{H}(\boldsymbol{\mu}) \le N_V$ and since $\hat{n}$ is a collection of positive functions, $\sum_{i,j \in [k]:i>j} \int_{\tilde{\Theta}_{ij} \cup \tilde{\Theta}_{ji}} \left( \sum_{s=1}^{k} \tilde{n}_s(\boldsymbol{\mu}) \right) \mathrm{d}\boldsymbol{H}(\boldsymbol{\mu}) \le N_V$, which means $\hat{n}$ satisfies the constraint of (16). By the minimality, $V_1^{\min} \le V_1(\hat{n})$, and since $\exp$ is a positive function, we have $V_1(\hat{n}) \le V(\hat{n}) = V^{\min}$.

Moreover, $V_1^{\min} \ge V_2^{\min}$ holds for

$$V_2^{\min} := \inf_{\tilde{n}: \mathbb{R}^k \to [0,\infty)^k} V_2(\tilde{n}) \tag{17}$$

$$\text{s.t. } \sum_{i \in [k]} \sum_{j \neq i} \int_{\tilde{\Theta}_{ij}} \left( \sum_{s=1}^{k} \tilde{n}_s(\boldsymbol{\mu}) \right) \mathrm{d}\boldsymbol{H}(\boldsymbol{\mu}) \le N_V$$

$$\text{where } V_2(\tilde{n}) := e^{-4} \sum_{i \in [k]} \sum_{j \neq i} \int_{\tilde{\Theta}_{ij}} \frac{\exp\left(-\left(\tilde{n}_i(\boldsymbol{\mu}) + \tilde{n}_j(\boldsymbol{\mu})\right) \frac{\tilde{\Delta}^2}{2 \min(\sigma_s^2)_{s \in [k]}}\right)}{8} \mathrm{d}H(\boldsymbol{\mu})$$

by using the fact that $\mathrm{KL}_i(\mu_i, \mu_j) = \frac{\tilde{\Delta}^2}{2\sigma_i^2} \le \frac{\tilde{\Delta}^2}{2 \min(\sigma_s^2)_{s \in [k]}}$.

**Claim 1.** We abuse our notation slightly so that $\boldsymbol{H}(E) = \int_E \mathrm{d}\boldsymbol{H}(\boldsymbol{\mu})$ for any Lebesgue measurable set $E$, and let $\tilde{\Theta} = \cup_{i,j \in [k]: i \neq j} \tilde{\Theta}_{ij}$ (note that all $\tilde{\Theta}_{ij}$ are mutually disjoint except for the measure zero sets). Then, the following $n^{opt}$ is an optimal solutions to (17)

$$n_s^{opt}(\boldsymbol{\mu}) := \frac{N_V}{2\boldsymbol{H}(\tilde{\Theta})} \mathbf{1} \Big[ \boldsymbol{\mu} \in \Big( \cup_{j \neq s} \tilde{\Theta}_{sj} \Big) \cup \Big( \cup_{i \neq s} \tilde{\Theta}_{is} \Big) \Big].$$

*Proof.* Choose an arbitrary $\tilde{n} : \mathbb{R}^k \to [0, \infty)^k$ which satisfies the constraint of optimization problem (17). Let $\tilde{N}(\boldsymbol{\mu}) := \sum_{s \in [k]} \tilde{n}_s(\boldsymbol{\mu})$. Now, since the function $\rho : x \mapsto \frac{1}{8R} \exp(-x \cdot \frac{\tilde{\Delta}^2}{2 \min(\sigma_s^2)_{s \in [k]}})$ is a convex and decreasing function, by Jensen's inequality we can say that

$$V_2(\tilde{n}) = \sum_{i \in [k]} \sum_{j \neq i} \int_{\tilde{\Theta}_{ij}} \rho\Big( \tilde{n}_i(\boldsymbol{\mu}) + \tilde{n}_j(\boldsymbol{\mu}) \Big) \mathrm{d}\boldsymbol{H}(\boldsymbol{\mu})$$

$$\geq \sum_{i \in [k]} \sum_{j \neq i} \boldsymbol{H}(\tilde{\Theta}_{ij}) \rho\Bigg( \frac{1}{\boldsymbol{H}(\tilde{\Theta}_{ij})} \int_{\tilde{\Theta}_{ij}} \Big( \tilde{n}_i(\boldsymbol{\mu}) + \tilde{n}_j(\boldsymbol{\mu}) \Big) \mathrm{d}\boldsymbol{H}(\boldsymbol{\mu}) \Bigg)$$

(Jensen's inequality for each integral on $\tilde{\Theta}_{ij}$)

$$= \boldsymbol{H}(\tilde{\Theta}) \cdot \sum_{i \in [k]} \sum_{j \neq i} \frac{\boldsymbol{H}(\tilde{\Theta}_{ij})}{\boldsymbol{H}(\tilde{\Theta})} \rho\Bigg( \frac{1}{\boldsymbol{H}(\tilde{\Theta}_{ij})} \int_{\tilde{\Theta}_{ij}} \Big( \tilde{n}_i(\boldsymbol{\mu}) + \tilde{n}_j(\boldsymbol{\mu}) \Big) \mathrm{d}\boldsymbol{H}(\boldsymbol{\mu}) \Bigg)$$

$$\geq \boldsymbol{H}(\tilde{\Theta}) \cdot \rho\Bigg( \sum_{i \in [k]} \sum_{j \neq i} \frac{\boldsymbol{H}(\tilde{\Theta}_{ij})}{\boldsymbol{H}(\tilde{\Theta})} \frac{1}{\boldsymbol{H}(\tilde{\Theta}_{ij})} \int_{\tilde{\Theta}_{ij}} \Big( \tilde{n}_i(\boldsymbol{\mu}) + \tilde{n}_j(\boldsymbol{\mu}) \Big) \mathrm{d}\boldsymbol{H}(\boldsymbol{\mu}) \Bigg)$$

(Jensen's inequality)

$$\geq \boldsymbol{H}(\tilde{\Theta}) \cdot \rho\Bigg( \sum_{i \in [k]} \sum_{j \neq i} \frac{1}{\boldsymbol{H}(\tilde{\Theta})} \int_{\tilde{\Theta}_{ij}} \Big( \sum_{s \in [k]} \tilde{n}_s \Big) \mathrm{d}\boldsymbol{H}(\boldsymbol{\mu}) \Bigg)$$

($\tilde{n}_i + \tilde{n}_j \leq \sum_{s \in [k]} \tilde{n}_s$ and $\rho$ is a decreasing function.)

$$\geq \boldsymbol{H}(\tilde{\Theta}) \cdot \rho\Bigg( \frac{N_V}{\boldsymbol{H}(\tilde{\Theta})} \Bigg).$$

(Constraint of (17))

Since $\tilde{n}$ is chosen arbitrarily, we can say $V_2^{\min} \geq \boldsymbol{H}(\tilde{\Theta})\rho\Big( \frac{N_V}{\boldsymbol{H}(\tilde{\Theta})} \Big)$. One can check the above $n^{opt}$ satisfies the constraint in the optimization problem (17) and also satisfies $V_2(n^{opt}) = \boldsymbol{H}(\tilde{\Theta})\rho(\frac{N_V}{\boldsymbol{H}(\tilde{\Theta})})$. Therefore, $n^{opt}$ is an optimal solution of optimization problem (17). $\square$

Using Lemma 9, we can get

$$H(\tilde{\Theta}) = \sum_{i \neq j} \int_{\Theta_{ij}} \mathbf{1}[|\mu_i - \mu_j| \leq \tilde{\Delta}] \mathbf{1}[|\mu_i|, |\mu_j| \leq \frac{1}{\sqrt{\tilde{\Delta}}}] \mathrm{d}\boldsymbol{H}(\boldsymbol{\mu}) = L_{ij}'(\boldsymbol{H}, \tilde{\Delta}). \tag{18}$$

Now applying Claim 1 and Eq. (18) on Optimization (17) implies the following result:

$$V_2^{\min} = V_2(n^{opt}) \tag{Claim 1}$$

$$= \frac{\exp\Big( -\frac{N_V}{2L'(\boldsymbol{H}, \tilde{\Delta})} \frac{\tilde{\Delta}^2}{2 \min(\sigma_s^2)_{s \in [k]}} \Big)}{8e^4} \sum_{i \neq j} \int_{\Theta_{ij}} \mathbf{1}[|\mu_i - \mu_j| \leq \tilde{\Delta}] \mathbf{1}[|\mu_i|, |\mu_j| \leq \frac{1}{\sqrt{\tilde{\Delta}}}] \mathrm{d}H(\boldsymbol{\mu})$$

(Eq. (18))

$$\geq \frac{\exp\Big( -\frac{N_V \tilde{\Delta}}{2 \min(\sigma_s^2)_{s \in [k]} L'(\boldsymbol{H}, \tilde{\Delta})} \Big)}{8e^4} \times L'(\boldsymbol{H}, \tilde{\Delta})\tilde{\Delta}. \tag{Eq. (18)}$$

If $V \leq \delta$ is true, then $V_2 \leq \delta$, which implies

$$\frac{\exp\left(-\frac{N_V \tilde{\Delta}}{2\min(\sigma_s^2)_{s\in[k]} L'(\boldsymbol{H},\tilde{\Delta})}\right)}{8e^4} \times L'(\boldsymbol{H},\tilde{\Delta}) \leq \delta \iff N_V \geq \frac{2\min(\sigma_s^2)_{s\in[k]} L'(\boldsymbol{H},\tilde{\Delta})}{\tilde{\Delta}} \ln \frac{L'(\boldsymbol{H},\tilde{\Delta})}{8e^4\delta}.$$

To make this lower bound greater than 0, $\ln \frac{L'(\boldsymbol{H},\tilde{\Delta})}{8e^4\delta} > 1$. From Lemma 9, we know that for small enough $\tilde{\Delta}$[9], $L'(\boldsymbol{H},\tilde{\Delta}) \in [\frac{1}{2}L(\boldsymbol{H}), 2L(\boldsymbol{H})]$. Setting

$$\tilde{\Delta} := \frac{32e^4}{L(\boldsymbol{H})}\delta, \tag{19}$$

we have

$$N_V \geq \min(\sigma_s^2)_{s\in[k]} \frac{L(\boldsymbol{H})^2}{16e^4} \ln 2 \tag{20}$$

which is the inequality we desired. $\qquad\square$

### D.1 Proof of Lemmas

#### D.1.1 Proof of lemma 11

*Proof.* It is equivalent to prove $p + q \geq \frac{1}{4}\exp(-d(p, 1 - q'))$ for any $p, q, q' \in (0, 1)$ such that $q' \leq q$. First, if $p \geq 1/4$ or $q \geq 1/4$ it trivially holds, and thus we assume $p < 1/4$ and $q < 1/4$. We have

$$d(p, 1 - q')d(p, 1 - q) \qquad\qquad (p, q \leq \tfrac{1}{4})$$
$$\leq d(p + q, 1 - (p + q)) \qquad\qquad (p + q < \tfrac{1}{2})$$
$$\leq \log \frac{1}{2.4(p + q)} \qquad\qquad \text{(Eq.(3) of Kaufmann et al. [2016a])}$$

and transforming this yields

$$p + q \geq \frac{1}{2.4} e^{-(d(p, 1 - q'))}.$$

This completes the proof. $\qquad\square$

#### D.1.2 Proof of the Lemma 12

*Proof.* We have

$$\frac{h_i(a)h_j(b)}{h_i(b)h_j(a)} = \exp\left(-\frac{(a - m_i)^2}{2\sigma_i^2} + \frac{(b - m_i)^2}{2\sigma_i^2} - \frac{(b - m_j)^2}{2\sigma_j^2} + \frac{(a - m_j)^2}{2\sigma_j^2}\right)$$

$$= \exp\left(-\frac{(a - b)(a + b - 2m_i)}{2\sigma_i^2} - \frac{(b - a)(a + b - 2m_j)}{2\sigma_j^2}\right)$$

$$= \exp\left(2(a - b)\left[\frac{m_i}{\sigma_i^2} - \frac{m_j}{\sigma_j^2}\right] - (a - b)(a + b)\left[\frac{1}{2\sigma_i^2} + \frac{1}{2\sigma_j^2}\right]\right)$$

$$\geq \exp\left(-2|a - b|\left|\frac{m_i}{\sigma_i^2} - \frac{m_j}{\sigma_j^2}\right| - |(a - b)(a + b)|\left[\frac{1}{2\sigma_i^2} + \frac{1}{2\sigma_j^2}\right]\right)$$

$$\geq \exp\left(-2\delta\left|\frac{m_i}{\sigma_i^2} - \frac{m_j}{\sigma_j^2}\right| - 2\sqrt{\delta}\left[\frac{1}{2\sigma_i^2} + \frac{1}{2\sigma_j^2}\right]\right)\frac{1}{R'_{ij}(\boldsymbol{H},\delta)}.$$

Define $R'(\boldsymbol{H},\delta) = \min_{i\neq j} R'_{ij}(\boldsymbol{H},\delta)$ which satisfies the condition of the Ratio lemma. For $\delta$ such that

$$\delta < D_1(\boldsymbol{H}) := \min_{i\neq j}\left[\left|\frac{m_i}{\sigma_i^2} - \frac{m_j}{\sigma_j^2}\right|^{-1}, \left[\frac{1}{2\sigma_i^2} + \frac{1}{2\sigma_j^2}\right]^{-2}\right],$$

---

[9] $\tilde{\Delta} < \min\left(D_0(\boldsymbol{H}), \min_{i\in[k]}\frac{1}{4m_i^2}, \frac{L(\boldsymbol{H})}{4(k-1)\sum_{i\in[k]}\frac{1}{\xi_i}}\right)$. Check Section G for the condition.

we have $R'(\boldsymbol{H}, \delta) \le e^4$.

$\square$

# E    Proof of Theorem 6

For notational convenience, define $\Delta_s(\boldsymbol{\mu}) := \mu_{i^*(\boldsymbol{\mu})} - \mu_s$ for $s \in [k]$, $\Delta(\boldsymbol{\mu}) := \min_{s \ne i^*(\boldsymbol{\mu})} \Delta_s = \Delta_{j^*(\boldsymbol{\mu})}$. Let $\mathcal{A}(t)$ be the subset of arms that have not been eliminated at time $t$.

Since $\delta < \frac{4L^2(\boldsymbol{H})}{\sum_{i \in [k]} \frac{k-1}{\xi_i}}$, $\Delta_0 = \frac{\delta}{4L(\boldsymbol{H})} \le \frac{L(\boldsymbol{H})}{\sum_{i \in [k]} \frac{k-1}{\xi_i}}$ and we have

$$\mathbb{P}_{\boldsymbol{\mu} \sim \boldsymbol{H}}(\Delta_0 \ge \Delta(\boldsymbol{\mu})) = \sum_{i \ne j} \int_{\Theta_{ij}} \mathbf{1}[|\mu_i - \mu_j| \le \Delta_0] \, \mathrm{d}\boldsymbol{H}(\boldsymbol{\mu})$$

$$= L(\boldsymbol{H}, \Delta_0)\Delta_0 \le 2L(\boldsymbol{H}) \cdot \Delta_0 \qquad \text{(Lemma 9)}$$

$$\le \frac{\delta}{2}. \qquad (21)$$

Next, we consider the upper bound estimator such that $\hat{\Delta}^{\mathrm{safe}}(t) \ge \Delta$ holds with high probability. Namely,

$$\mathrm{UCB}(i, t), \mathrm{LCB}(i, t) = \hat{\mu}_i(t) \pm \mathrm{Conf}(i, t),$$

$$\mathrm{Conf}(i, t) = \sqrt{2\sigma_i^2 \frac{\log(6(N_i(t))^2/((\frac{\delta^2}{2k})\pi^2))}{N_i(t)}},$$

$$\hat{\Delta}^{\mathrm{safe}}(t) = \max_i \mathrm{UCB}(i, t) - \max_j \mathrm{LCB}(j, t).$$

From the definition above, we can calculate how many arm pulls the learner needs to narrow down the confidence width.

**Lemma 13.** Define $B := 320 \max_{i \in [k]} \sigma_i^2$ and $\Delta_{thr} := \min\left(\log \frac{4\sqrt{k}}{\delta\pi}, \frac{1}{B}\right)$. Let $R_0(\Delta) := B \frac{\log \min(\Delta, \Delta_{thr})^{-1}}{\min(\Delta, \Delta_{thr})^2}$. Then, for any $\Delta \in (0, \infty)$ and for a timestep $t$ which satisfies $N_i(t) \ge R_0(\Delta)$, $\mathrm{Conf}(i, t) \le \Delta/4$.

*Proof of Lemma 13.* Only for this part of the proof, let $\Delta' := \min(\Delta, \Delta_{thr})$ for notational convenience. Then,

$$\mathrm{Conf}(i, t) = \sqrt{2\sigma_i^2 \frac{\log(6(N_i(t))^2/((\frac{\delta^2}{2k})\pi^2))}{N_i(t)}}$$

$$\le \sqrt{2\sigma_i^2 \frac{\log(6R_0(\Delta)^2/((\frac{\delta^2}{2k})\pi^2))}{R_0(\Delta)}} \qquad \text{(assumption on } t\text{)}$$

$$= \Delta' \sqrt{2\sigma_i^2} \sqrt{\frac{\log(6(B \times \frac{\log(\Delta'^{-1})}{\Delta'^2})^2/((\frac{\delta^2}{2k})\pi^2))}{B \times \log(\Delta'^{-1})}}$$

$$\le \Delta' \sqrt{2\sigma_i^2} \sqrt{\frac{\log(6(\frac{B}{\Delta'^3})^2/((\frac{\delta^2}{2k})\pi^2))}{B\log(\Delta'^{-1})}} \qquad (\log \Delta'^{-1} \le \Delta'^{-1})$$

$$= \Delta' \sqrt{2\sigma_i^2} \sqrt{\frac{6\log(\Delta'^{-1}) + 2\log B + \log \frac{12k}{\delta^2\pi^2}}{B\log(\Delta'^{-1})}}$$

$$\le \Delta' \sqrt{2\sigma_i^2} \sqrt{\frac{6}{B} + \frac{2}{B} + \frac{2}{B}} \qquad \text{(Definition of } \Delta_{thr}, \Delta_{thr} \ge \Delta')$$

$$= \Delta' \sqrt{\frac{20\sigma_i^2}{B}} \le \frac{1}{4}\Delta' \le \frac{1}{4}\Delta.$$

$\square$

From this lemma, one could induce the following corollary which states that Algorithm 5 always terminates before a certain timestep:

**Corollary 14.** Let $\tau$ (and $\gamma$) be the stopping time (and the last iteration of the while loop in Algorithm 1, respectively) where Algorithm 1 meets the stopping condition. Then, $\gamma$ is always bounded by $R_0(\Delta_0)$ and $\tau$ is uniformly bounded by $T_0 := k \cdot R_0(\Delta_0)$.

*Proof of Corollary 14.* Let us assume that $\tau > T_0 + 1$. Then, by Lemma 13, each $i \in \mathcal{A}_{T_0}$ satisfies $\mathrm{Conf}(i, T_0) \leq \Delta_0/4$. Let $i^{ucb}(t) = \arg\max_{i \in \mathcal{A}(t)} \mathrm{UCB}(i, t)$ and $i^{lcb}(t) = \arg\max_{i \in \mathcal{A}(t)} \mathrm{LCB}(i, t)$. From definition,

$$
\begin{aligned}
\hat{\Delta}^{\mathrm{safe}}(T_0) &= \max_{i \in \mathcal{A}(T_0)} \mathrm{UCB}(i, T_0) - \max_{i \in \mathcal{A}(T_0)} \mathrm{LCB}(i, T_0) \\
&= \mathrm{UCB}(i^{ucb}(T_0), T_0) - \mathrm{LCB}(i^{lcb}(T_0), T_0) \\
&= 2\mathrm{Conf}(i^{ucb}(T_0), T_0) + 2\mathrm{Conf}(i^{lcb}(T_0), T_0) + \mathrm{LCB}(i^{ucb}(T_0), T_0) - \mathrm{UCB}(i^{lcb}(T_0), T_0) \\
&\leq \Delta_0 + \mathrm{LCB}(i^{ucb}(T_0), T_0) - \mathrm{UCB}(i^{lcb}(T_0), T_0) \\
&\leq \Delta_0 \qquad\qquad\text{(Since both arms survived from the elimination phase)}
\end{aligned}
$$

which implies $\hat{\Delta}^{\mathrm{safe}} \leq \Delta_0$, contradicting $\tau \geq T_0$ since the algorithm should be terminated by Line 18 at timestep $T_0$. Therefore, $\tau \leq T_0$. $\qquad\square$

Lemma 14 implies Algorithm 1 always stops before $T_0$ samples. Morever, the following lemma states that with high probability, true mean $\boldsymbol{\mu}$ is in between UCB and LCB for all time steps.

**Lemma 15.** (Uniform confidence bound) The following holds for all $i \in [k]$:

$$
\mathbb{P}_{\boldsymbol{\mu}}\left[\bigcap_{t=1}^{\infty} \{\mathrm{LCB}(i, t) \leq \mu_i\}\right] \geq 1 - \frac{\delta^2}{2k}, \tag{22}
$$

$$
\mathbb{P}_{\boldsymbol{\mu}}\left[\bigcap_{t=1}^{\infty} \{\mu_i \leq \mathrm{UCB}(i, t)\}\right] \geq 1 - \frac{\delta^2}{2k}. \tag{23}
$$

*Proof of Lemma 15.* The following derives the upper bound part, Eq. (23). The lower bound is derived by following the same steps.

Since each arm is independent of each other, Eq. (23) boils down to prove

$$
\mathbb{P}_{\boldsymbol{\mu}}\left[\bigcap_{s=1}^{\infty} \{\mu_i \leq \mathrm{UCB}(i, t_i(s))\}\right] \geq 1 - \frac{\delta^2}{2k},
$$

where $t_i(s) = \min\{t \in \mathbb{N} : N_i(t) \geq s\}$ and $t_i(s) = \infty$ if $\{t \in \mathbb{N} : N_i(t) \geq s\} = \emptyset$. For each event $\{\mu_i \leq UCB_i(t_i(s))\}$, since each arm pull is independent of each other, by Hoeffding's inequality we have

$$
\mathbb{P}_{\boldsymbol{\mu}}\left(\frac{1}{s}\sum_{j=1}^{s}(X_j^i - \mu_i) \leq -\epsilon\right) \leq \exp\left(-\frac{s\epsilon^2}{2\sigma_0^2}\right)
$$

for any $\epsilon > 0$. If we set $\epsilon = \mathrm{Conf}(i, t_i(s))$, we can transform the above inequality to

$$
\mathbb{P}_{\boldsymbol{\mu}}\big(\mathrm{UCB}(i, t_i(s)) \leq \mu_i\big) \leq \frac{\delta^2}{2ks^2} \cdot \frac{6}{\pi^2}.
$$

By the union bound,

$$
\mathbb{P}_{\boldsymbol{\mu}}\left[\bigcap_{s=1}^{\infty} \{\mu_i \leq \mathrm{UCB}(i, t_i(s))\}\right] \geq 1 - \sum_{s=1}^{\infty} \mathbb{P}_{\boldsymbol{\mu}}\big[\mu_i \geq \mathrm{UCB}(i, t_i(s))\big]
$$

$$\geq 1 - \sum_{s=1}^{\infty} \frac{\delta^2}{2ks^2} \cdot \frac{6}{\pi^2}$$

$$\geq 1 - \sum_{s=1}^{\infty} \frac{\delta^2}{2ks^2} \cdot \frac{6}{\pi^2} = 1 - \frac{\delta^2}{2k},$$

and the proof is completed. □

Let us define a good event based on Lemma 15 as

$$\mathcal{X}(\boldsymbol{\mu}) := \bigcap_{i \in [k]} \left[ \left( \bigcap_{t=1}^{\infty} \{\mathrm{LCB}(i,t) \leq \mu_i\} \right) \bigcap \left( \bigcap_{t=1}^{\infty} \{\mathrm{UCB}(i,t) \geq \mu_i\} \right) \right]. \tag{24}$$

We now prove that, under $\mathcal{H}_{\boldsymbol{\mu}}$ and under this good event $\mathcal{X}(\boldsymbol{\mu})$

- The best arm $i^*(\boldsymbol{\mu})$ is always in the active arm set $\mathcal{A}(t)$ for all $t$ (Lemma 16),

- Each count of the suboptimal arm pull, $N_i(T_0)$, is bounded by roughly $O(\frac{\log \Delta_i}{\Delta_i^2})$ (Lemma 17).

**Lemma 16.** Let

$$\mathcal{X}'(\boldsymbol{\mu}) = \bigcap_{t=1}^{\infty} \{i^*(\boldsymbol{\mu}) \in \mathcal{A}(t)\}.$$

Then, under $\mathcal{H}_{\boldsymbol{\mu}}$, $\mathcal{X}(\boldsymbol{\mu}) \subset \mathcal{X}'(\boldsymbol{\mu})$ and therefore

$$\mathbb{P}_{\boldsymbol{\mu}}\left[(\mathcal{X}'(\boldsymbol{\mu}))^c\right] \leq \delta^2.$$

*Proof of Lemma 16.* Suppose that event $\mathcal{X}(\boldsymbol{\mu})$ occurs under $\mathcal{H}_{\boldsymbol{\mu}}$. Then for all $i \in [k]$ and for all $t$, $\hat{\mu}_i - \mathrm{Conf}(i,t) \leq \mu_i$ and $\hat{\mu}_i + \mathrm{Conf}(i,t) \geq \mu_i$. Now, for any $i \neq i^*$,

$$\begin{aligned}
\mathrm{LCB}(i,t) - \mathrm{UCB}(i^*,t) &= \hat{\mu}_i - \mathrm{Conf}(i,t) - (\hat{\mu}_{i^*} + \mathrm{Conf}(i^*,t)) \\
&\leq \mu_i + \mathrm{Conf}(i,t) - \mathrm{Conf}(i,t) - (\mu_{i^*} - \mathrm{Conf}(i^*,t) + \mathrm{Conf}(i^*,t)) \\
&\hspace{8cm} \text{(Event } \mathcal{X} \text{ occurs)} \\
&= \mu_i - \mu_{i^*} < 0
\end{aligned}$$

which means when event $\mathcal{X}$ occurs, the optimal arm will never be dropped, and thus $\mathcal{X} \subset \mathcal{X}'$. By Lemma 15,

$$\mathbb{P}_{\boldsymbol{\mu}}(\mathcal{X}'(\boldsymbol{\mu})) \geq \mathbb{P}_{\boldsymbol{\mu}}(\mathcal{X}(\boldsymbol{\mu})) \geq 1 - \delta^2.$$

□

**Lemma 17.** For any $i \neq i^*$, under $\mathcal{H}_{\boldsymbol{\mu}}$ we have

$$\left\{N_i(T_0) > R_0\left(\max(\Delta_i, \Delta_0)\right)\right\} \subset \mathcal{X}(\boldsymbol{\mu})^c, \tag{25}$$

and therefore, $\mathbb{E}_{\boldsymbol{\mu}}[N_i(T_0)\mathbf{1}[\mathcal{X}(\boldsymbol{\mu})]] \leq R_0\left(\max(\Delta_i, \Delta_0)\right)$.

*Proof of Lemma 17.* Only for this part of the proof, let $T_i := R_0\left(\max(\Delta_i, \Delta_0)\right)$ for brevity.

When $\Delta_i < \Delta_0$, $\max(\Delta_i, \Delta_0) = \Delta_0$, which, combined with Corollary 14, implies that $\{N_i(T_0) \geq T_i\}$ always holds.

For the case of $\Delta_i > \Delta_0$, suppose that the learner is under the events $\mathcal{X}(\boldsymbol{\mu})$ and $\{i \in \mathcal{A}(T_i)\}$. Note that

$$\mathrm{Conf}(a, T_i) \leq \frac{\Delta_i}{4}, \quad \forall a \in \mathcal{A}(T_i).$$

Then,

$$\max_{a \in \mathcal{A}(T_i)} \mathrm{LCB}_a(T_i) - \mathrm{UCB}_i(T_i) \geq \mathrm{LCB}_{i^*}(T_i) - \mathrm{UCB}_i(T_i)$$

$$\begin{aligned}
&= \hat{\mu}_{i^*} - \mathrm{Conf}_{i^*}(T_i) - (\hat{\mu}_i + \mathrm{Conf}_i(T_i)) \\
&\geq \mu_{i^*} - 2\mathrm{Conf}_{i^*}(T_i) - (\mu_i + 2\mathrm{Conf}_i(T_i)) \qquad (\mathcal{X} \text{ occurs}) \\
&\geq \mu_{i^*} - \mu_i - \Delta_i = 0.
\end{aligned}$$

Therefore, when $\mathcal{X}$ occurs, $\mu_i$ should be eliminated after timestep $T_i$ so $N_i(T_0) \leq T_i$. $\qquad \square$

Lemmas 16 and 17 guarantee the Bayesian $\delta$-correctness of our Algorithm 1.

**Theorem 18.** ($\delta$-correctness) The Bayesian PoE of Algorithm 1 is at most $\delta$, i.e.,

$$\int_{\mathbb{R}^k} \mathrm{PoE}(\boldsymbol{\mu})\, \mathrm{d}\boldsymbol{H}(\boldsymbol{\mu}) \leq \delta. \tag{26}$$

*Proof of Theorem 18.* Throughout the proof, we use the following results.

- The probability that $\mu_{i^*(\boldsymbol{\mu})} - \mu_{j^*(\boldsymbol{\mu})} \leq \Delta_0$ is at most $\frac{\delta}{2}$ by Eq. (21).

- The event $\mathcal{X}$ fails to hold with probability at most $\delta^2$ by Lemma 15.

- When $\mu_{i^*(\boldsymbol{\mu})} - \mu_{j^*(\boldsymbol{\mu})} > \Delta_0$ and event $\mathcal{X}$ occurs, by Lemmas 16 and 17, all suboptimal arms will be eliminated before $T_{j^*}$ and only the optimal arm will remain in the set. This means $\mathcal{E}(\boldsymbol{\mu}) \subset \mathcal{X}(\boldsymbol{\mu}) \cup \{\mu_{i^*(\boldsymbol{\mu})} - \mu_{j^*(\boldsymbol{\mu})} \leq \Delta_0\}$.

Therefore,

$$\begin{aligned}
\mathrm{PoE}(\pi; \boldsymbol{H}) &= \mathbb{E}_{\boldsymbol{\mu}\sim\boldsymbol{H}}\left[\mathbb{P}\Big(J \neq i^*(\boldsymbol{\mu})|\mathcal{H}_{\boldsymbol{\mu}}\Big)\right] = \mathbb{E}_{\boldsymbol{\mu}\sim\boldsymbol{H}}\left[\mathbb{E}[\mathbf{1}(\mathcal{E}(\boldsymbol{\mu}))|\mathcal{H}_{\boldsymbol{\mu}}]\right] \\
&\leq \mathbb{E}_{\boldsymbol{\mu}\sim\boldsymbol{H}}\left[\mathbb{E}[\mathbf{1}(\{\mu_{i^*(\boldsymbol{\mu})} - \mu_{j^*(\boldsymbol{\mu})} \leq \Delta_0\}) + \mathbf{1}(\mathcal{X}(\boldsymbol{\mu}))|\mathcal{H}_{\boldsymbol{\mu}}]\right] \\
&\leq \mathbb{E}_{\boldsymbol{\mu}\sim\boldsymbol{H}}\left[\mathbf{1}(\{\mu_{i^*(\boldsymbol{\mu})} - \mu_{j^*(\boldsymbol{\mu})} \leq \Delta_0\}) + \mathbb{E}[\mathbf{1}(\mathcal{X}(\boldsymbol{\mu}))|\mathcal{H}_{\boldsymbol{\mu}}]\right] \\
&\leq \frac{\delta}{2} + \delta^2 < \delta.
\end{aligned}$$

$\qquad \square$

Finally, Lemma 19 shows the upper bound of the expected stopping time of our algorithm.

**Lemma 19.** We have $\mathbb{E}[\tau] \leq B_0 \frac{L(\boldsymbol{H})^2}{\delta} \log\left(\frac{L(\boldsymbol{H})}{\delta}\right) + O(\log \delta^{-1})$.

*Proof.* We have

$$\begin{aligned}
\mathbb{E}[\tau] &= \sum_{s=1}^{k} \mathbb{E}[N_s(T_0)] = \sum_{s=1}^{k} \mathbb{E}[\mathbb{E}_{\boldsymbol{\mu}}[N_s(T_0)]] \\
&= \sum_{s=1}^{k} \mathbb{E}\left[\mathbb{E}_{\boldsymbol{\mu}}[N_s(T_0)\mathbf{1}[\mathcal{X}(\boldsymbol{\mu})]]\right] + \sum_{s=1}^{k} \mathbb{E}\left[\mathbb{E}_{\boldsymbol{\mu}}[N_s(T_0)\mathbf{1}[\mathcal{X}^c(\boldsymbol{\mu})]]\right] \\
&\leq \sum_{s=1}^{k} \mathbb{E}\left[\mathbb{E}_{\boldsymbol{\mu}}[N_s(T_0)\mathbf{1}[\mathcal{X}(\boldsymbol{\mu})]]\right] + T_0 \cdot \delta^2 \qquad \text{(Corollary 14 and Lemma 15)}
\end{aligned}$$

and since $\Delta_0 < \Delta_{thr}$ by assumption,

$$T_0 \cdot \delta^2 = k \cdot R_0(\Delta_0) \cdot \delta^2 \leq kB \cdot \frac{\log \Delta_0^{-1}}{\Delta_0^2} \cdot \delta^2. \tag{27}$$

Therefore, it remains to compute the scale of the first term, $\sum_{s=1}^{k} \mathbb{E}\left[\mathbb{E}_{\boldsymbol{\mu}}[N_s(T_0)\mathbf{1}[\mathcal{X}(\boldsymbol{\mu})]]\right]$.

The following evaluates $\mathbb{E}\Big[\mathbb{E}_{\boldsymbol{\mu}}[N_i(T_0)\mathbf{1}[\mathcal{X}(\boldsymbol{\mu})]]\Big]$ for each $i$. For notational convenience, let $\mathcal{T}_s(\boldsymbol{\mu}) :=$ $\mathbb{E}_{\boldsymbol{\mu}}\Big[R_0(\max(\Delta_s,\Delta_0))\mathbf{1}[\mathcal{X}(\boldsymbol{\mu})]\Big]$. Then,

$$\mathbb{E}\Big[\mathbb{E}_{\boldsymbol{\mu}}[N_s(T_0)\mathbf{1}[\mathcal{X}(\boldsymbol{\mu})]]\Big] \leq \mathbb{E}[\mathcal{T}_s(\boldsymbol{\mu})] \tag{Lemma 17}$$

$$= \sum_{i=1}^{k} \mathbb{E}[\mathcal{T}_s(\boldsymbol{\mu})\mathbf{1}[\boldsymbol{\mu} \in \Theta_i]]$$

$$= \sum_{i=1}^{k} \int_{\mu \in \Theta_i} \mathcal{T}_s(\boldsymbol{\mu})\,\mathrm{d}\boldsymbol{H}(\boldsymbol{\mu}) \tag{28}$$

(Recall that $\Theta_i = \{\boldsymbol{\mu} \in \mathbb{R}^k : \mu_i \geq \max_{j \neq i}\mu_j\}$). In the following, we calculate each $\int_{\mu \in \Theta_i}\mathcal{T}_s(\boldsymbol{\mu})\,\mathrm{d}\boldsymbol{H}(\boldsymbol{\mu})$. By assumption, $\Delta_0 < \Delta_{thr}$.

For this part, we define a new notation that for each vector $v \in \mathbb{R}^k$ and $i,j \in [k]$, $v_{\backslash i}$ (and $v_{\backslash i,j}$) is the projection of $v$ to $\mathbb{R}^{k-1}$ ($\mathbb{R}^{k-2}$, respectively) by omitting $i$-th coordinate ($i,j$-th coordinate, respectively), $v_{\backslash i} := \underbrace{(v_1, v_2, \cdots, v_{i-1}, v_{i+1}, \cdots, v_k)}_{k-1 \text{ coordinates}}$. Similarly, for a $k$-dimensional distribution $\boldsymbol{H}$ and $i,j \in [k]$, $\boldsymbol{H}_{\backslash i}$ (and $\boldsymbol{H}_{\backslash i,j}$) is the distribution which omits $i$-th ($i,j$-th, respectively) coordinate.

**Case 1:** $s \neq i$   In this case, by Lemma 17,

$$\int_{\mu \in \Theta_i} \mathcal{T}_s(\boldsymbol{\mu})\,\mathrm{d}\boldsymbol{H}(\boldsymbol{\mu}) = \int_{\mu \in \Theta_i} \mathcal{T}_s(\boldsymbol{\mu})(\mathbf{1}[\mu_i - \mu_s \geq \Delta_0] + \mathbf{1}[\mu_i - \mu_s < \Delta_0])\,\mathrm{d}\boldsymbol{H}(\boldsymbol{\mu})$$

$$= \int_{\mu_i \in \mathbb{R}} \int_{\mu_s = -\infty}^{\mu_i - \Delta_{thr}} \int_{\mu_{\backslash i,s} \in (-\infty, \mu_i)^k} \mathcal{T}_s(\boldsymbol{\mu})h_i(\mu_i)h_s(\mu_s)\,\mathrm{d}\boldsymbol{H}_{\backslash i,s}\,\mathrm{d}\mu_s\,\mathrm{d}\mu_i$$

$$+ \int_{\mu_i \in \mathbb{R}} \int_{\mu_s = \mu_i - \Delta_{thr}}^{\mu_i - \Delta_0} \int_{\mu_{\backslash i,s} \in (-\infty, \mu_i)^k} \mathcal{T}_s(\boldsymbol{\mu})h_i(\mu_i)h_s(\mu_s)\,\mathrm{d}\boldsymbol{H}_{\backslash i,s}\,\mathrm{d}\mu_s\,\mathrm{d}\mu_i$$

$$+ \int_{\mu_i \in \mathbb{R}} \int_{\mu_s = \mu_i - \Delta_0}^{\mu_i} \int_{\mu_{\backslash i,s} \in (-\infty, \mu_i)^k} \mathcal{T}_s(\boldsymbol{\mu})h_i(\mu_i)h_s(\mu_s)\,\mathrm{d}\boldsymbol{H}_{\backslash i,s}\,\mathrm{d}\mu_s\,\mathrm{d}\mu_i$$

$$\leq B \cdot \frac{\log \Delta_{thr}^{-1}}{\Delta_{thr}^2}$$

$$+ \int_{\mu_i \in \mathbb{R}} \int_{\mu_s = \mu_i - \Delta_{thr}}^{\mu_i - \Delta_0} \int_{\mu_{\backslash i,s} \in (-\infty, \mu_i)^k} B\frac{\log(\mu_i - \mu_s)^{-1}}{(\mu_i - \mu_s)^2}h_i(\mu_i)h_s(\mu_s)\,\mathrm{d}\boldsymbol{H}_{\backslash i,s}\,\mathrm{d}\mu_s\,\mathrm{d}\mu_i$$

$$+ \int_{\mu_i \in \mathbb{R}} \int_{\mu_s = \mu_i - \Delta_0}^{\mu_i} \int_{\mu_{\backslash i,s} \in (-\infty, \mu_i)^k} B\frac{\log \Delta_0^{-1}}{\Delta_0^2}h_i(\mu_i)h_s(\mu_s)\,\mathrm{d}\boldsymbol{H}_{\backslash i,s}\,\mathrm{d}\mu_s\,\mathrm{d}\mu_i$$

$$\tag{Lemma 17}$$

$$= \int_{\mu_i \in \mathbb{R}} \int_{\mu_s = -\mu_i - \Delta_{thr}}^{\mu_i - \Delta_0} B\frac{\log(\mu_i - \mu_s)^{-1}}{(\mu_i - \mu_s)^2}h_i(\mu_i)h_s(\mu_s)\left[\prod_{k \in [k]\backslash\{i,s\}} H_k(\mu_i)\right]\mathrm{d}\mu_s\,\mathrm{d}\mu_i$$

$$+ B\frac{\log \Delta_0^{-1}}{\Delta_0^2}\mathbb{P}(i^*(\mu) = i, \mu_i - \mu_s \leq \Delta_0) + B \cdot \frac{\log \Delta_{thr}^{-1}}{\Delta_{thr}^2}$$

$$\leq B\log \Delta_0^{-1} \underbrace{\int_{\mu_i \in \mathbb{R}} h_i(\mu_i) \int_{\mu_s = -\infty}^{\mu_i - \Delta_0} \frac{1}{(\mu_i - \mu_s)^2}h_s(\mu_s)\left[\prod_{k \in [k]\backslash\{i,s\}} H_k(\mu_i)\right]\mathrm{d}\mu_s\,\mathrm{d}\mu_i}_{(A)}$$

$$+ B\frac{\log \Delta_0^{-1}}{\Delta_0^2} \underbrace{\mathbb{P}(i^*(\mu) = i, \mu_i - \mu_s \le \Delta_0)}_{(P_{is})} + B\frac{\log \Delta_{thr}^{-1}}{\Delta_{thr}^2}.$$

We first deal with the term $(A)$. The following splits $(A)$ into the sum of two integrals $(A1)$ and $(A2)$:

$$(A) = \int_{\mu_i \in \mathbb{R}} h_i(\mu_i) \left[ \prod_{k \in [k] \setminus \{i, s\}} H_k(\mu_i) \right] \int_{\mu_s = -\infty}^{\mu_i - \Delta_0} \frac{1}{(\mu_i - \mu_s)^2} h_s(\mu_s) \, \mathrm{d}\mu_s \, \mathrm{d}\mu_i$$

$$= \underbrace{\sum_{l=1}^{\lceil \frac{1}{2\sqrt{\Delta_0}} \rceil - 1} \int_{\mu_i \in \mathbb{R}} h_i(\mu_i) \left[ \prod_{k \in [k] \setminus \{i, s\}} H_k(\mu_i) \right] \int_{\mu_s = \mu_i - (l+1)\Delta_0}^{\mu_i - l\Delta_0} \frac{1}{(\mu_i - \mu_s)^2} h_s(\mu_s) \, \mathrm{d}\mu_s \, \mathrm{d}\mu_i}_{(A1)}$$

$$+ \underbrace{\int_{\mu_i \in \mathbb{R}} h_i(\mu_i) \left[ \prod_{k \in [k] \setminus \{i, s\}} H_k(\mu_i) \right] \int_{\mu_s = -\infty}^{\mu_i - \lceil \frac{1}{2\sqrt{\Delta_0}} \rceil \Delta_0} \frac{1}{(\mu_i - \mu_s)^2} h_s(\mu_s) \, \mathrm{d}\mu_s \, \mathrm{d}\mu_i}_{(A2)}.$$

For $(A1)$, we have

$$(A1) \le \sum_{l=1}^{\lceil \frac{1}{2\sqrt{\Delta_0}} \rceil - 1} \int_{\mu_i \in \mathbb{R}} h_i(\mu_i) \left[ \prod_{k \in [k] \setminus \{i, s\}} H_k(\mu_i) \right] \int_{\mu_s = \mu_i - (l+1)\Delta_0}^{\mu_i - l\Delta_0} \frac{1}{l^2 \Delta_0^2} h_s(\mu_s) \, \mathrm{d}\mu_s \, \mathrm{d}\mu_i$$

$$= \sum_{l=1}^{\lceil \frac{1}{2\sqrt{\Delta_0}} \rceil - 1} \frac{1}{l^2 \Delta_0^2} \mathbb{P}(i^*(\mu) = i, \mu_i - \mu_s \in [l\Delta_0, (l+1)\Delta_0])$$

$$\le \sum_{l=1}^{\lceil \frac{1}{2\sqrt{\Delta_0}} \rceil - 1} \frac{1}{l^2 \Delta_0^2} \cdot 2\mathbb{P}(i^*(\mu) = i, \mu_i - \mu_s \le \Delta_0) \qquad \text{(Lemma 20)}$$

$$\le \sum_{l=1}^{\infty} \frac{1}{l^2 \Delta_0^2} \cdot 2P_{is}$$

$$= \frac{\pi^2}{3\Delta_0^2} P_{is}. \tag{29}$$

Now for $(A2)$, we evaluate the inner integral of $(A2)$:

$$(\text{Inner} - A2) := \frac{1}{\sqrt{2\pi}\sigma_s} \int_{-\infty}^{\mu_i - \lceil \frac{1}{2\sqrt{\Delta_0}} \rceil \Delta_0} \frac{1}{(\mu_i - \mu_s)^2} \exp\left( -\frac{(\mu_s - m_s)^2}{2\sigma_s^2} \right) \, \mathrm{d}\mu_s$$

$$= \frac{1}{\sqrt{2\pi}\sigma_s} \left[ \frac{1}{(\mu_i - \mu_s)} \exp\left( -\frac{(\mu_s - m_s)^2}{2\sigma_s^2} \right) \right]_{-\infty}^{\mu_i - \lceil \frac{1}{2\sqrt{\Delta_0}} \rceil \Delta_0}$$

$$+ \frac{1}{\sqrt{2\pi}\sigma_s} \int_{-\infty}^{\mu_i - \lceil \frac{1}{2\sqrt{\Delta_0}} \rceil \Delta_0} \frac{\mu_s - m_s}{\sigma_s^2(\mu_i - \mu_s)} \exp\left( -\frac{(\mu_s - m_s)^2}{2\sigma_s^2} \right) \, \mathrm{d}\mu_s$$

$$\text{(partial integration)}$$

$$\le \frac{1}{\sqrt{2\pi}\sigma_s} \frac{1}{\lceil \frac{1}{2\sqrt{\Delta_0}} \rceil \Delta_0} + \frac{1}{\sqrt{2\pi}\sigma_s^3} \int_{-\infty}^{\mu_i - \lceil \frac{1}{2\sqrt{\Delta_0}} \rceil \Delta_0} \left( \frac{\mu_i - m_s}{\mu_i - \mu_s} - 1 \right) \exp\left( -\frac{(\mu_s - m_s)^2}{2\sigma_s^2} \right) \, \mathrm{d}\mu_s$$

$$(\mu_i > \mu_s)$$

$$\leq \frac{1}{\sqrt{2\pi}\sigma_s} \frac{1}{\lceil \frac{1}{2\sqrt{\Delta_0}}\rceil \Delta_0} + \frac{1}{\sqrt{2\pi}\sigma_s^3} \int_{-\infty}^{\mu_i - \lceil \frac{1}{2\sqrt{\Delta_0}}\rceil \Delta_0} \frac{\mu_i - m_s}{\lceil \frac{1}{2\sqrt{\Delta_0}}\rceil \Delta_0} \exp\left(-\frac{(\mu_s - m_s)^2}{2\sigma_s^2}\right) d\mu_s$$

$$\leq \frac{1}{\sqrt{2\pi}\sigma_s} \frac{1}{\lceil \frac{1}{2\sqrt{\Delta_0}}\rceil \Delta_0} + \max\left[\frac{1}{\sigma_s^2} \frac{\mu_i - m_s}{\lceil \frac{1}{2\sqrt{\Delta_0}}\rceil \Delta_0}, 0\right].$$

By integrating above over variable $i$, we have

$$(A2) \leq \int_{\mathbb{R}} h_i(\mu_i) \left[\prod_{k\in[k]\backslash\{i,s\}} H_k(\mu_i)\right] \left(\frac{1}{\sqrt{2\pi}\sigma_s} \frac{1}{\lceil \frac{1}{2\sqrt{\Delta_0}}\rceil \Delta_0} + \max\left[\frac{1}{\sigma_s^2} \frac{\mu_i - m_s}{\lceil \frac{1}{2\sqrt{\Delta_0}}\rceil \Delta_0}, 0\right]\right) d\mu_i$$

$$\leq \int_{\mathbb{R}} h_i(\mu_i) \left(\frac{1}{\sqrt{2\pi}\sigma_s} \frac{1}{\lceil \frac{1}{2\sqrt{\Delta_0}}\rceil \Delta_0} + \max\left[\frac{1}{\sigma_s^2} \frac{\mu_i - m_s}{\lceil \frac{1}{2\sqrt{\Delta_0}}\rceil \Delta_0}, 0\right]\right) d\mu_i \qquad (F_k(\cdot) \leq 1)$$

$$= \frac{1}{\sqrt{2\pi}\sigma_s} \frac{1}{\lceil \frac{1}{2\sqrt{\Delta_0}}\rceil \Delta_0} + \frac{1}{\sigma_s^2 \lceil \frac{1}{2\sqrt{\Delta_0}}\rceil \Delta_0} \left[\int_{\mathbb{R}} h_i(\mu_i) \max[\mu_i - m_s, 0] \, d\mu_i\right]$$

$$\leq \frac{1}{\sqrt{2\pi}\sigma_s} \frac{1}{\lceil \frac{1}{2\sqrt{\Delta_0}}\rceil \Delta_0} + \frac{1}{\sigma_s^2 \lceil \frac{1}{2\sqrt{\Delta_0}}\rceil \Delta_0} \left[\int_{\mathbb{R}} h_i(\mu_i) |\mu_i - m_s| \, d\mu_i\right]$$

$$\leq \frac{1}{\sqrt{2\pi}\sigma_s} \frac{1}{\lceil \frac{1}{2\sqrt{\Delta_0}}\rceil \Delta_0} + \frac{1}{\sigma_s^2 \lceil \frac{1}{2\sqrt{\Delta_0}}\rceil \Delta_0} \left[\int_{\mathbb{R}} h_i(\mu_i) \big(|\mu_i - m_i| + |m_i - m_s|\big) \, d\mu_i\right]$$

$$\leq \frac{1}{\sqrt{2\pi}\sigma_s} \frac{1}{\lceil \frac{1}{2\sqrt{\Delta_0}}\rceil \Delta_0} + \frac{1}{\sigma_s^2 \lceil \frac{1}{2\sqrt{\Delta_0}}\rceil \Delta_0} \left(|m_i - m_s| + \frac{\sigma_i \sqrt{2}}{\sqrt{\pi}}\right)$$

$$\text{(Mean of Half normal distribution is } \frac{\sigma_i \sqrt{2}}{\sqrt{\pi}})$$

$$= \frac{1}{\lceil \frac{1}{2\sqrt{\Delta_0}}\rceil \Delta_0} \underbrace{\left[\frac{1}{\sqrt{2\pi}\sigma_s} + \frac{1}{\sigma_s^2}\left(|m_i - m_s| + \frac{\sigma_i \sqrt{2}}{\sqrt{\pi}}\right)\right]}_{S_{is}(\boldsymbol{H})/2} \leq \frac{S_{is}(\boldsymbol{H})}{\sqrt{\Delta_0}} = O(\Delta_0^{-1/2}). \qquad (30)$$

Therefore, by Eq. (29) and Eq. (30),

$$(A) = (A1) + (A2) = \frac{1}{\Delta_0^2} \cdot \frac{\pi^2}{3} P_{is} + O(\Delta_0^{-1/2}),$$

and therefore

$$\int_{\mu\in\Theta_i} \mathcal{T}_s(\boldsymbol{\mu}) \, d\boldsymbol{H}(\boldsymbol{\mu}) \leq \frac{B \log \Delta_0^{-1}}{\Delta_0^2} P_{is} \left[\frac{\pi^2}{3} + 1\right] + O(\Delta_0^{-1/2}). \qquad (31)$$

**Case 2:** $s = i$    In this case, let

$$\int_{\mu\in\Theta_s} \mathcal{T}_s(\boldsymbol{\mu}) \, d\boldsymbol{H}(\boldsymbol{\mu}) = \sum_{j\neq s} \int_{\mu\in\Theta_{sj}} \mathcal{T}_s(\boldsymbol{\mu}) \, d\boldsymbol{H}(\boldsymbol{\mu})$$

$$\leq \sum_{j\neq s} \int_{\mu\in\Theta_{sj}} \max_{i\neq s} (\mathcal{T}_i(\boldsymbol{\mu})) \, d\boldsymbol{H}(\boldsymbol{\mu})$$

$$(N_s(t) \text{ increases only when } |\mathcal{A}_t| > 1, \text{ so there should be a competitor})$$

$$\leq \sum_{j\neq s} \int_{\mu\in\Theta_{sj}} \max_{i\neq s} \big[R_0(\Delta_i(\boldsymbol{\mu}), \Delta_0)\big] \, d\boldsymbol{H}(\boldsymbol{\mu}) \qquad \text{(Lemma 17)}$$

$$\leq \sum_{j\neq s} \int_{\mu\in\Theta_{sj}} \big[R_0(\Delta_j(\boldsymbol{\mu}), \Delta_0)\big] \, d\boldsymbol{H}(\boldsymbol{\mu})$$

and by the same calculation as Case 1, we obtain

$$\int_{\mu \in \Theta_{sj}} \left[ R_0(\Delta_j(\boldsymbol{\mu}), \Delta_0) \right] \mathrm{d}\boldsymbol{\mu} \le \frac{B \log \Delta_0^{-1}}{\Delta_0^2} P_{sj} \left[ \frac{\pi^2}{3} + 1 \right] + O(\Delta_0^{-1/2}),$$

and therefore

$$\int_{\mu \in \Theta_s} \mathcal{T}_s(\boldsymbol{\mu}) \, \mathrm{d}\boldsymbol{H}(\boldsymbol{\mu}) \le \frac{B \log \Delta_0^{-1}}{\Delta_0^2} \left[ \frac{\pi^2}{3} + 1 \right] \sum_{j \ne s} P_{sj} + O(\Delta_0^{-1/2}). \tag{32}$$

For notational convenience, let $B_0 = B \cdot \left[ \frac{\pi^2}{3} + 1 \right]$. From Eq. (28), Eq. (31), Eq. (32), we get

$$\mathbb{E}[N_s(T_0)] = \mathbb{E}_{\boldsymbol{\mu} \sim \boldsymbol{H}} \left[ \mathbb{E}_{\boldsymbol{\mu}}[N_s(T_0) \mathbf{1}[\mathcal{X}(\boldsymbol{\mu})]] \right] + \mathbb{E}_{\boldsymbol{\mu} \sim \boldsymbol{H}} \left[ \mathbb{E}_{\boldsymbol{\mu}}[N_s(T_0) \mathbf{1}[\mathcal{X}(\boldsymbol{\mu})^c]] \right] \tag{33}$$

$$\le \sum_{i=1}^{k} \int_{\mu \in \Theta_i} \mathcal{T}_s(\boldsymbol{\mu}) \, \mathrm{d}\boldsymbol{H}(\boldsymbol{\mu}) + \mathbb{E}_{\boldsymbol{\mu} \sim \boldsymbol{H}} \left[ \mathbb{E}_{\boldsymbol{\mu}}[N_s(T_0) \mathbf{1}[\mathcal{X}(\boldsymbol{\mu})^c]] \right] \qquad \text{(Eq. (28))}$$

$$\le \sum_{i \ne s} \int_{\mu \in \Theta_i} \mathcal{T}_s(\boldsymbol{\mu}) \, \mathrm{d}\boldsymbol{H}(\boldsymbol{\mu}) + \int_{\mu \in \Theta_s} \mathcal{T}_s(\boldsymbol{\mu}) \, \mathrm{d}\boldsymbol{H}(\boldsymbol{\mu}) + \mathbb{E}_{\boldsymbol{\mu} \sim \boldsymbol{H}} \left[ \mathbb{E}_{\boldsymbol{\mu}}[N_s(T_0) \mathbf{1}[\mathcal{X}(\boldsymbol{\mu})^c]] \right]$$

$$\le \frac{B_0 \log \Delta_0^{-1}}{\Delta_0^2} \left[ \sum_{i \ne s} P_{is} + \sum_{j \ne s} P_{sj} \right] + O(\Delta_0^{-1/2}) + \mathbb{E}_{\boldsymbol{\mu} \sim \boldsymbol{H}} \left[ \mathbb{E}_{\boldsymbol{\mu}}[N_s(T_0) \mathbf{1}[\mathcal{X}(\boldsymbol{\mu})^c]] \right]$$

$$\text{(Eq. (31) and (32))}$$

$$= \frac{B_0 \log \Delta_0^{-1}}{\Delta_0^2} \left[ \mathbb{P}(i^*(\boldsymbol{\mu}) \ne s, \mu_{i^*(\boldsymbol{\mu})} - \mu_s \le \Delta_0) + \mathbb{P}(i^*(\boldsymbol{\mu}) = s, \mu_s - \mu_{j^*(\boldsymbol{\mu})} \le \Delta_0) \right]$$

$$+ O(\Delta_0^{-1/2}) + \mathbb{E}_{\boldsymbol{\mu} \sim \boldsymbol{H}} \left[ \mathbb{E}_{\boldsymbol{\mu}}[N_s(T_0) \mathbf{1}[\mathcal{X}(\boldsymbol{\mu})^c]] \right]. \tag{34}$$

Now, the total stopping time is bounded as follows:

$$\mathbb{E}[\tau] = \mathbb{E}[\sum_{s=1}^{k} N_s(T_0)]$$

$$= \frac{B_0 \log \Delta_0^{-1}}{\Delta_0^2} \left[ \underbrace{\sum_{s=1}^{k} \mathbb{P}(i^*(\boldsymbol{\mu}) \ne s, \mu_{i^*(\boldsymbol{\mu})} - \mu_s \le \Delta_0)}_{\text{Psum1}} + \underbrace{\sum_{s=1}^{k} \mathbb{P}(i^*(\boldsymbol{\mu}) = s, \mu_s - \mu_{j^*(\boldsymbol{\mu})} \le \Delta_0)}_{\text{Psum2}} \right]$$

$$+ O(\Delta_0^{-1/2}) + \mathbb{E}[\tau \mathbf{1}[\mathcal{X}^c]]. \qquad \text{(Eq. (34))}$$

The final task we have left is bounding (Psum1) and (Psum2). Let us define $k^*(\boldsymbol{\mu})$ as the third best arm in $\boldsymbol{\mu}$. For (Psum1),

$$(\text{Psum1}) = \sum_{s=1}^{k} \mathbb{P}(i^*(\boldsymbol{\mu}) \ne s, \mu_{i^*(\boldsymbol{\mu})} - \mu_s \le \Delta_0)$$

$$= \sum_{s=1}^{k} \mathbb{P}(j^*(\boldsymbol{\mu}) = s, \mu_{i^*(\boldsymbol{\mu})} - \mu_s \le \Delta_0) + \sum_{s=1}^{k} \mathbb{P}(j^*(\boldsymbol{\mu}) \ne s, \mu_{i^*(\boldsymbol{\mu})} - \mu_s \le \Delta_0)$$

$$\le \mathbb{P}(\mu_{i^*(\boldsymbol{\mu})} - \mu_{j^*(\boldsymbol{\mu})} \le \Delta_0) + \sum_{s=1}^{k} \mathbb{P}(\mu_{i^*(\boldsymbol{\mu})} - \mu_{k^*(\boldsymbol{\mu})} \le \Delta_0)$$

$$\text{(When } j^*(\boldsymbol{\mu}) \ne s, \mu_{k^*(\boldsymbol{\mu})} \ge s)$$

$$\leq \frac{\delta}{2} + k \cdot \mathbb{P}(\mu_{i^*(\boldsymbol{\mu})} - \mu_{k^*(\boldsymbol{\mu})} \leq \Delta_0) \qquad \text{(Definition of } \Delta_0)$$

$$\leq \frac{\delta}{2} + O(\Delta_0^2). \qquad \text{(Lemma 21)}$$

For (Psum2),

$$\text{(Psum2)} = \sum_{s=1}^{k} \mathbb{P}(i^*(\boldsymbol{\mu}) = s, \mu_s - \mu_{j^*(\boldsymbol{\mu})} \leq \Delta_0)$$

$$= \mathbb{P}(\mu_{i^*(\boldsymbol{\mu})} - \mu_{j^*(\boldsymbol{\mu})} \leq \Delta_0)$$

$$= \mathbb{P}(\Delta(\boldsymbol{\mu}) \leq \Delta_0) \leq \frac{\delta}{2} \qquad \text{(Definition of } \Delta_0)$$

and finally we can conclude

$$\mathbb{E}[\tau] \leq \frac{B_0 \log \Delta_0^{-1}}{\Delta_0^2}\delta + O(\Delta_0^{-1/2}) + \mathbb{E}[\tau\mathbf{1}[\mathcal{X}^c]] \qquad \text{(Eq. (34), (Psum1) and (Psum2))}$$

$$\leq \frac{B_0 \log \Delta_0^{-1}}{\Delta_0^2}\delta + O(\Delta_0^{-1/2}) + \frac{KB \log \Delta_0^{-1}}{\Delta_0^2} \cdot \delta^2 \qquad \text{(Eq.(27))}$$

$$= \frac{B_0 \log \Delta_0^{-1}}{\Delta_0^2}\delta + O(\Delta_0^{-1/2}), \qquad (35)$$

and the proof is completed. □

### E.1 Bound that holds for any $\delta$

In this case, we need to change the definition of $\Delta_0$ as

$$\Delta_0 := \min\left(\max\left\{\Delta \in (0,1) : L(\boldsymbol{H}, \Delta)\Delta \leq \frac{\delta}{2}\right\}, \min_{i \neq j}(\xi_i L_{ij}(\boldsymbol{H}))^2\right). \qquad (36)$$

Assume that $\Delta_0 < \Delta_{thr}$. Then, all the proof flows of Section E after Eq. (21) follows accordingly, and we obtain the following upper bound:

$$\mathbb{E}[\tau] \leq \underbrace{\frac{B_0 \log \Delta_0^{-1}}{\Delta_0^2}\delta + \frac{2\sum_{i \neq j} S_{ij}(\boldsymbol{H})}{\sqrt{\Delta_0}}}_{(35)} + \underbrace{\frac{KB_0 \log \Delta_0^{-1}}{\Delta_0^2}\delta^2}_{(27)} + \underbrace{B_0\left(\sum_{i \neq j \neq k} Q_{ijk}\right)\log \Delta_0^{-1}}_{\text{From (Psum1)}} + k^2 \cdot \frac{B_0 \log \Delta_{thr}^{-1}}{\Delta_{thr}^2},$$

$$(37)$$

where $S_{ij}$ and $Q_{ijk}$ are defined in Eq. (30) and Lemma 21, respectively.

Eq. (37) also holds for the case of $\Delta_0 \geq \Delta_{thr}$. In this case, from the definition of $R_0$, we have

$$\mathcal{T}_s(\boldsymbol{\mu}) = \mathbb{E}_{\boldsymbol{\mu}}\Big[R_0(\max(\Delta_s, \Delta_0))\mathbf{1}[\mathcal{X}]\Big] = \mathbb{E}_{\boldsymbol{\mu}}\big[R_0(\Delta_{thr})\mathbf{1}[\mathcal{X}]\big] \leq R_0(\Delta_{thr})\mathbb{P}_{\boldsymbol{\mu}}[\mathcal{X}] = R_0(\Delta_{thr}),$$

which implies

$$\mathbb{E}[\tau] \leq k \cdot R_0(\Delta_{thr}) + T_0 \cdot \delta^2$$

$$= k \cdot R_0(\Delta_{thr}) \cdot (1 + \delta^2) \qquad (T_0 = k \cdot R_0(\Delta_0) = R_0(\Delta_{thr}))$$

$$= k \cdot \frac{B_0 \log \Delta_{thr}^{-1}}{\Delta_{thr}^2} \cdot (1 + \delta^2)$$

$$\leq 2k \frac{B_0 \log \Delta_{thr}^{-1}}{\Delta_{thr}^2} \leq \text{Eq. (37).}$$

In summary, we obtain Eq. (37).

### E.2 Proof of Lemmas

#### E.2.1 Proof of Lemma 20

**Lemma 20.** For $S + 1 \leq \frac{1}{\sqrt{\delta}}$ and for $\delta \leq \left(\xi_i L_{is}(\boldsymbol{H})\right)^2$, we have

$$\mathbb{P}(i^*(\boldsymbol{\mu}) = i, \mu_i - \mu_s \in [S\delta, (S+1)\delta]) \leq 2\mathbb{P}(i^*(\boldsymbol{\mu}) = i, \mu_i - \mu_s \leq \delta).$$

*Proof.* We have

$$\int_{\Theta_i} \mathbf{1}\left[|\mu_i - \mu_s| \in [S\delta, (S+1)\delta]\right] \mathrm{d}\boldsymbol{H}(\boldsymbol{\mu}) = \int_{\Theta_{\backslash i}} \int_{\mu_i = \mu_s + S\delta}^{\mu_s + (S+1)\delta} h_i(\mu_i)\, \mathrm{d}\mu_i\, \mathrm{d}\boldsymbol{H}_{\backslash i}(\boldsymbol{\mu}_{\backslash i})$$

$$\leq \int_{\Theta_{\backslash i}} \delta\left[h_i(\mu_s) + \frac{e^{-1/2}}{\xi_i}(S+1)\delta\right] \mathrm{d}\boldsymbol{H}_{\backslash i}(\boldsymbol{\mu}_{\backslash i})$$

(by Lipschitz property of Gaussian, $e^{-1/2}/\xi_i$ is the steepest slope of $N(m_i, \xi_i^2)$)

$$= \delta\Big(\underbrace{\left[\int_{\Theta_{\backslash i}} h_i(\mu_s)\, \mathrm{d}\boldsymbol{H}(\boldsymbol{\mu}_{\backslash i})\right]}_{L_{is}(\boldsymbol{H})} + \frac{e^{-1/2}}{\xi_i}(S+1)\delta\Big)$$

$$\leq (L_{is}(\boldsymbol{H}) + \frac{1}{\xi_i}(S+1)\delta)\delta.$$

When $S + 1 < \frac{1}{\sqrt{\delta}}$ and $\sqrt{\delta} < L_{is}(\boldsymbol{H})\xi_i$, we have

$$\int_{\Theta_i} \mathbf{1}\left[|\mu_i - \mu_j| \in [S\delta, (S+1)\delta]\right] \mathrm{d}\boldsymbol{H}(\boldsymbol{\mu}) \leq 2L_{is}(\boldsymbol{H})\delta$$

as intended. $\qquad \square$

#### E.2.2 Proof of Lemma 21

We will show that the probability that three or more arms are $\delta$-close is $O(\delta^2)$. Namely:

**Lemma 21.** We have $\mathbb{P}(\mu_{i^*(\boldsymbol{\mu})} - \mu_{k^*(\boldsymbol{\mu})} \leq \Delta_0) = O(\delta^2)$.

*Proof.* For any $i \neq j \neq k \in [k]$, we have

$$\int \mathbf{1}\left[|\mu_i - \mu_j|, |\mu_i - \mu_k| \leq \delta\right] \mathrm{d}\boldsymbol{H}(\boldsymbol{\mu})$$

$$= \int_{\boldsymbol{\mu}_{\backslash jk}} \int_{\mu_j = \mu_i - \delta}^{\mu_i + \delta} \int_{\mu_k = \mu_i - \delta}^{\mu_i + \delta} h_j(\mu_j) h_k(\mu_k)\, \mathrm{d}\mu_k\, \mathrm{d}\mu_j\, \mathrm{d}\boldsymbol{H}_{\backslash jk}(\boldsymbol{\mu}_{\backslash jk})$$

$$\leq \int_{\boldsymbol{\mu}_{\backslash jk}} \int_{\mu_j = \mu_i - \delta}^{\mu_i + \delta} \int_{\mu_k = \mu_i - \delta}^{\mu_i + \delta} \left(h_j(\mu_i) + \frac{e^{-1/2}\delta}{\xi_j}\right)\left(h_k(\mu_i) + \frac{e^{-1/2}\delta}{\xi_k}\right) \mathrm{d}\mu_k\, \mathrm{d}\mu_j\, \mathrm{d}\boldsymbol{H}_{\backslash jk}(\boldsymbol{\mu}_{\backslash jk})$$

(Lipschitz property of Gaussian)

$$\leq (2\delta)^2 \underbrace{\int_{\boldsymbol{\mu}_{\backslash jk}} \left[h_j(\mu_i) h_k(\mu_i) + O(\delta)\right] \mathrm{d}\boldsymbol{H}_{\backslash jk}(\boldsymbol{\mu}_{\backslash jk})}_{=:Q_{ijk}(\boldsymbol{H})} = O(\delta^2).$$

$\qquad \square$

## F  Experimental details

The code used in the experiments for this paper can be found at the following link: `https://github.com/jajajang/FC_BAI_Bayes`.

## F.1 Stopping condition

**TTTS** We use our theoretical results stated in Section 5 for our stopping criterion. For TTTS, we use Chernoff's stopping rule, as Garivier and Kaufmann [2016], Jourdan et al. [2022] did. Here is the description of how it works: for each arm $i, j \in [k]$, let

$$\hat{\mu}_{ij}(t) := \frac{N_i(t)}{N_i(t) + N_j(t)} \hat{\mu}_i(t) + \frac{N_j(t)}{N_i(t) + N_j(t)} \hat{\mu}_j(t),$$

and define

$$Z_{ij}(t) := N_i(t) \cdot KL_i(\hat{\mu}_i, \hat{\mu}_{ij}) + N_j(t) \cdot KL_j(\hat{\mu}_j, \hat{\mu}_{ij}).$$

Now the stopping time is defined as:

$$\tau_{TTTS} := \inf\{t \in \mathbb{N} : \max_{a \in [k]} \min_{b \in [k] \setminus a} Z_{ab}(t) \geq \beta(t, \delta)\}$$

for some threshold function $\beta(t, \delta)$, which is defined by the following proposition of Garivier and Kaufmann [2016]:

**Theorem 22** (Garivier and Kaufmann 2016, Proposition 12)**.** Let $\boldsymbol{\mu}$ be an exponential family bandit model. Let $\delta \in (0, 1)$ and $\alpha > 1$. There exists a constant $C = C(\alpha, k)$ such that whatever the sampling strategy, using Chernoff's stopping rule with the threshold

$$\beta(t, \delta) = \log \frac{Ct^\alpha}{\delta}$$

ensures that for all $\boldsymbol{\mu}$, $\mathbb{P}_{\boldsymbol{\mu}}(\tau < \infty, J \neq i^*(\boldsymbol{\mu})) \leq \delta$.

In this theorem, we give an advantage to the stopping time of TTTS by setting $\alpha = 1, C = 1$. Theoretically, $C(\alpha, k) > 1$ and $C \to \infty$ as $\alpha \to 1_+$, but we set the threshold smaller than the theoretical guarantee so that TTTS stops earlier.

**TTUCB** We followed the stopping rule of the original paper Jourdan and Degenne [2022b]. Let $\mathcal{C}_G(x) := \min_{\lambda \in (\frac{1}{2}, 1]} \frac{2\lambda - 2\lambda \log(4\lambda) + \log \zeta(2\lambda) - 0.5 \log(1-\lambda) + x}{\lambda}$ where $\zeta$ is a Riemann $\zeta$ function, and

$$c(n, \delta) := 2\mathcal{C}_G(\frac{1}{2} \log \frac{k-1}{\delta}) + 4 \log(4 + \log \frac{n}{2}).$$

Let $\hat{i}_t := \arg\max_{i \in [k]} \hat{\mu}_i(t)$, the empirical best arm at step $t$. The TTUCB algorithm stops when

$$\min_{i \neq \hat{i}_t} \frac{\hat{\mu}_{\hat{i}}(t) - \hat{\mu}_i(t)}{\sqrt{\frac{1}{N_{\hat{i}_t}(t)} + \frac{1}{N_i(t)}}} \geq \sqrt{c(t, \delta)}.$$

When the algorithm stops sampling, the TTUCB algorithm recommends the empirical best arm as its final suggestion.

Since the computation of $\mathcal{C}_G(x)$ involves optimization, it is computationally heavy when the number of samples is excessively large (as our Table 1). Instead, we approximated $\mathcal{C}_G(x) \approx x + \log x$ as mentioned in Jourdan and Degenne [2022b].

## F.2 NoElim algorithm

The NoElim algorithm is shown in Algorithm 2.

## F.3 Tables including computation time

For all tables in this section, Comp represents the average computation time (second).

**Algorithm 2** No Elimination (NoElim) Algorithm
___
**Input:** Confidence level $\delta$, prior $\boldsymbol{H}$
$\Delta_0 := \frac{\delta}{4L(\boldsymbol{H})}$
Initialize the candidate of best arms $\mathcal{A}(1) = [k]$
$t = 1$
**while** True **do**
  ***Draw each arm in*** $[k]$ ***once.*** {Main Difference with Algorithm 1}.
  $t \to t + |\mathcal{A}(t)| \, \hat{\Delta}^{\text{safe}}(t)$.
  **for** $i \in \mathcal{A}(t)$ **do**
    Calculate UCB$(i,t)$ and LCB$(i,t)$ from (5).
    **if** UCB$(i,t) \le \max_j$ LCB$(j,t)$ **then**
      $\mathcal{A}(t) \leftarrow \mathcal{A}(t) \setminus \{i\}$.
    **end if**
  **end for**
  **if** $|\mathcal{A}(t)| = 1$ **then**
    **Return** arm $J$ in $\mathcal{A}(t)$.
  **end if**
  Calculate safe empirical gap
  **if** $\hat{\Delta}^{\text{safe}}(t) \le \Delta_0$ **then**
    **Return** arm $J$ which is uniformly sampled from $\mathcal{A}(t)$.
  **end if**
**end while**
___

Table 6: Extended version of Table 1 with computation time.

|  | AVG | MAX | ERROR | COMP. |
|---|---|---|---|---|
| ALG. 1 | $1.06 \times 10^4$ | $2.35 \times 10^5$ | 1.5% | 0.17 |
| TTTS | $1.56 \times 10^5$ | $1.09 \times 10^8$ | 0.5% | 27.6 |
| TTUCB | $1.95 \times 10^5$ | $1.13 \times 10^8$ | 0% | 5.07 |

Table 7: Extended version of Table 2 with computation time.

|  | AVG | MAX | ERROR | COMP. |
|---|---|---|---|---|
| ALG. 1 | $2.69 \times 10^5$ | $1.66 \times 10^7$ | 0.6% | 1.59 |
| NOELIM | $1.29 \times 10^6$ | $8.25 \times 10^7$ | 0% | 5.5 |

## F.4 Additional experiment results - Multiple arms, different prior mean/variance

In Section 6, we only used $k = 2$ arms for the simulation of Table 1. We made a brief comparison between Algorithm 1 and TTUCB in $k = 10$ arm environment with a prior distribution where prior means and variances are all different.

- Number of arms $K = 10$

- Prior means: we sample 10 random numbers from $N(0,1)$ before the experiment starts, and set them as prior means. Here is the list of prior means: [-0.053 0.528 -0.332 -0.368 -0.273 0.909 0.418 -1.17 0.873 -0.405]

- Prior variance: we sample 10 random numbers from $\mathsf{Unif}([0.5, 1.5])$ before the experiment starts, and set them as prior means. Here is the list of prior std: [0.604 1.477 1.163 0.988 0.560 0.513 0.997 1.332 0.828 0.833]

- Instance variance: we sample 10 random numbers from $\mathsf{Unif}([0.5, 1.5])$ before the experiment starts, and set them as prior means. Here is the list of instance std: [1.498, 1.262, 1.485, 0.963, 1.375, 0.969, 1.357, 1.238, 1.088, 0.699]

- Number of experiments: 500

- We stop additional sampling of TTUCB when its number of samples is over $10^8$ because of the time constraint. This means we gave some 'advantage' on TTUCB about expected stopping time (since it makes TTUCB stop earlier than it should.)

**Result**  Our algorithm had the average stopping time of $1.1 \times 10^5$, while TTUCB had the average stopping time of $7.1 \times 10^5$, again proving the superiority of Algorithm 1 over TTUCB in Bayesian settings.

Table 8: Multiple arms.

| | AVG | MAX | ERROR | COMP. |
|---|---|---|---|---|
| ALG. 1 | $1.1 \times 10^5$ | $2.58 \times 10^6$ | 1.5% | 1.52 |
| TTUCB | $7.1 \times 10^5$ | $10^8$(CAPPED) | 0.1% | 6.22 |

### F.5  Miscellaneous

**Computation of $\Delta_0$**  From Definition 2,

$$L_{ij}(\boldsymbol{H}) := \int_{-\infty}^{\infty} h_i(x) h_j(x) \prod_{s:s \in [k] \setminus \{i,j\}} H_s(x) \, \mathrm{d}x.$$

In **Scipy** package, there are functions for computing the cumulative function of Gaussian $H_s$ (scipy.norm.cdf) and $h_i$ (scipy.norm.pdf). **Scipy** package also supports the numerical integration (scipy.integrate.quad) which we use to numerically compute $L_{ij}$ in our experiments.

**Codes**  The codes are in the following GitHub repository: `https://github.com/jajajang/FC_BAI_Bayes`.

**Hardware**  We used Python 3.7 as our programming language and Macbook Pro M2 16 inch as our hardware.

## G  Scale restriction on $\delta$ for each theorem

The second result of the Lemma 9 is used for the lower bound. For this result to hold, we need the following two conditions for $D_1$:

- Proof of Lemma 1, second result: For the proof of Eq. (9), we used $|m_i| \leq \frac{1}{2\sqrt{\Delta}}$.

- Proof of Lemma 1, second result: For the proof of Eq. (9), we used $\frac{1}{\xi_i} \Delta^2 > 2 \exp\left(-\frac{1}{8\Delta \xi_i^2}\right) + 2 \exp\left(-\frac{1}{8\Delta \xi_j^2}\right)$. To satisfy this condition, $\Delta < D_0(\boldsymbol{H})$ where

$$D_0(\boldsymbol{H}) := \begin{cases} W\left(-\frac{1}{32 \max_{i \in [k]} \xi_i^{3/2}}\right) & \text{If } \max_{i \in [k]} \xi_i > \sqrt[3]{\frac{e^2}{2^{10}}} \\ 1 & \text{Otherwise} \end{cases}.$$

Here $W$ is the Lambert W function with the principal branch.

For the lower bound proof, we consider sufficiently small $\delta$ subject to the following constraints on $\tilde{\Delta} = \frac{32e^4}{L(\boldsymbol{H})} \delta$:

- Two conditions above for Lemma 9.

- For the Lemma 12: $\tilde{\Delta} < D_1(\boldsymbol{H}) := \min_{i \neq j}\left[\left|\frac{m_i}{\sigma_i^2} - \frac{m_j}{\sigma_j^2}\right|^{-1}, \left[\frac{1}{2\sigma_i^2} + \frac{1}{2\sigma_j^2}\right]^{-2}\right]$.

- To make $L'(\boldsymbol{H}, \tilde{\Delta})\tilde{\Delta} \in (\frac{1}{2}L(\boldsymbol{H}), 2L(\boldsymbol{H}))$, $\tilde{\Delta} \leq \frac{L(\boldsymbol{H})}{4\sum_{i \in [k]} \frac{k-1}{\xi_i}}$.

In summary, Theorem 1 holds for any

$$\delta \le \delta_L := \frac{L(\boldsymbol{H})}{32e^4} \cdot \min\left( D_0(\boldsymbol{H}), D_1(\boldsymbol{H}), \min_{i \in [k]} \frac{1}{4m_i^2}, \frac{L(\boldsymbol{H})}{4(k-1)\sum_{i \in [k]} \frac{1}{\xi_i}} \right).$$

For the upper bound proof (Theorem 6), we consider $\delta$ such that $\Delta_0 = \frac{\delta}{4L(\boldsymbol{H})}$ satisfies the following conditions:

- $\Delta_0 < \frac{L(\boldsymbol{H})}{\sum_{i \in [k]} \frac{k-1}{\xi_i}}$ to make $L(\boldsymbol{H}, \Delta_0) \cdot \Delta_0 \le 2L(\boldsymbol{H})\Delta_0$ by the first result of Lemma 9.

- $\Delta_0 \le \min_{i,j \in [k], i \ne j}(L_{ij}\xi_i)^2$ for the proof and usage of Lemma 20, and

- $\Delta_0 < \Delta_{thr}$.

