# OpenReview forum: "Fixed Confidence Best Arm Identification in the Bayesian Setting"
_NeurIPS.cc/2024/Conference — NeurIPS 2024 poster_

### Official Review · Reviewer_4HmL · 2024-06-15

**Soundness:** 3
**Presentation:** 4
**Contribution:** 2
**Rating:** 5
**Confidence:** 3

**Summary:**

This paper considers the best arm identification (BAI) problem in the fixed confidence setting and in the Bayesian setting, where the mean rewards of each arm is drawn from a known prior distribution. The authors formulate the problem and discuss the related literature. The authors provide the lower bound of the expected sample complexity of finding the optimal arm, which is proportional $L(\mathbb{H})/\delta$ where $L(\mathbb{H})$ is defined by the authors to characterize the sample complexity. Then the authors show the current optimal algorithms in the frequentist setting: the top two and tack-and-stop algorithms are suboptimal in the Bayesian setting. The authors then propose a new algorithm based on the successive elimination algorithm in the frequentist setting, showing that this algorithm reaches the optimal sample complexity up to a logarithmic factor. The authors provide the upper bound of the algorithm on the expected sample complexity. Finally the authors run two simulations to show that the top-two algorithm is indeed suboptimal comparing with Algorithm 1, and also show the superior performance of algorithm1 compared to the no-elimination version of it.

**Strengths:**

1. The problem is clearly defined and the paper is nicely presented.
2. It is nice to show that the optimal algorithms (top-two and tack-and-stop) are suboptimal in the Bayesian setting
3. The authors illustrate the theorems and proofs using a simpler 2-arm version which can be better understood.

**Weaknesses:**

1. The assumption of the arms may be stringent, i.e. all arms are from the Gaussian distribution. The current literature considers many more general distributions, such as sub-gaussian, exponential family, or even non-parametric cases.
2. The confidence interval is not optimized to the tightest confidence interval which is suggested in the lil-UCB paper from Jamieson: "lil' UCB : An Optimal Exploration Algorithm for Multi-Armed Bandits"
3. The lower bound proof (and possibly the upper bound proof) uses standard techniques from the literature. Perhaps the authors could point out the difficulty or challenge in the proofs.
4. The upper bound is not optimal compared to the lower bound, with an extra logarithmic factor. Is it possible to reduce this gap using tighter confidence bound or do some modification to the algorithm, such as modification of top-two or tack-and-stop instead of modify the SE algorithm (which is not optimal in the frequentist setting)
5. The simulations are limited to just two sets of experiments. The baselines or compared algorithms are just 2/1 in the respective setting. And it might be better to include different $\delta$ values since the authors consider the fixed confidence setting and change priors as well.

**Questions:**

1. Is it possible to use tighter confidence intervals to get a better upper bound?
2. Is it trivial or direct to consider similar algorithms when the distribution of the arms is more general, like subgaussian, or non-parametric?
3. Is it possible to modify top-two or tack-and-stop to make it better in the Bayesian setting?
4. Just out of curiosity, is it possible to modify LUCB or lil-UCB to this setting so that it achieves similar performance when the authors modify the successive elimination algorithm? Since the former two algorithms have better sample complexity in the frequentist setting.
5. Can the upper bound be modified to a high probability bound instead of an expectation bound on the sample complexity? Would it be challenging or trivial?

**Limitations:**

See weaknesses and questions.

---

> ### Author Rebuttal · Authors · 2024-08-05
>
> We thank the reviewer for the valuable review and comments. The following are the responses to the questions you raised.
>
> > Q1) better upper bound using lil-UCB
>
> We assume your question refers to the improvement of the confidence bound from $\sqrt{\log(N_i(t)/\delta)/N_i(t)}$ to $\sqrt{\log(\log(N_i(t))/\delta)/N_i(t)}$. It is challenging to verify all parts of the proof during this short rebuttal period, but we conjecture this can improve the dependency on $\log(L(H))$ but would not change the dependency with respect to $\log(1/\delta)$. Since this is the first paper on the fixed-confidence Bayesian best arm identification, we consider the current analysis to have enough novelty.
>
> > Q2) extending the distribution of the arms is more general, like subgaussian, or non-parametric?
>
> Many results in our paper, such as Lemma 1 (Volume lemma), Lemmas 11, 13~17, and Theorem 18 hold also in non-Gaussian settings. However, several results such as Lemmas 12 and 17 are tailored for the Gaussian setting and we cannot extend those results easily to other environments. Therefore, our lower and upper bound results need some additional work to extend to other settings. As we mentioned in Section 7, extending the current results to more diverse environments is one of the interesting future research topics.
>
> > Q3, Q4) modify other algorithms, such as top-two, track-and-stop, LUCB or lil-UCB to make it better in the Bayesian setting
>
> As we have mentioned in our future work section (Section 7), it is another interesting research topic. We chose the Elimination algorithm since it is easier to manage the number of suboptimal arm pulls. Elimination also attains an orderwise optimal ratio in a frequentist setting, so we believe that using TT, T&S, LUCB or lil-UCB won't make an orderwise improvement, especially with respect to $\delta$.
>
> > Q5) a high probability bound instead of an expectation bound on the sample complexity?
>
> In the Bayesian setting in general, there is no natural idea of a high-probability bound, since the bandit instance $\mu$ is also another random variable. For example, in many FC-BAI studies in a frequentist setting, they propose the high-probability bounds of the stopping time as the form of **'$\tau \leq f(\mu)$ with probability $1-\delta$'**. It is reasonable in a frequentist setting since $\mu$ is a fixed instance in their setting. However, in our case, $\mu$ is also a random variable, so it means now the bold inequality compares two random variables. We thought that whether such results could be accepted as high-probability bounds might be controversial depending on the reader. Therefore, we presented expectation bounds.
>
> In the case of the high-probability bound when the bandit instance $\mu$ is fixed, it is relatively trivial. Our Lemma 17 can be seen as the high probability bound for each arm when $\mu$ is given, and from this result we computed the overall expectation, which was computationally challenging (about 5 pages of integral computations).
>
> > W3) The lower bound proof (and possibly the upper bound proof) uses standard techniques from the literature.
>
> Our lower bound proof is novel and we suggest the first framework to prove Bayesian FC-BAI lower bound. Our proof structure is significantly different from the standard technique in the Frequentist FC-BAI lower bound, such as Kaufmann et al., (2016).
>
> The main differences come from the different definitions of the error probability (PoE) between the frequentist setting and the Bayesian setting. As mentioned in Lines 178-179, Bayesian $\delta$-correctness is more lenient than the frequentist $\delta$-correctness. This means we need to consider more diverse algorithms for the lower bound since the lower bound is for all possible $\delta$-correct algorithms. We want to mention three challenges in our proof.
>
> - First, in our case, we suggest a novel interpretation of PoE to an optimization problem (Opt0 -> Opt1, given the upper bound of the stopping time, how small the error probability can be?)
>
> - Second, we propose a relaxation (Opt1 -> Opt2), which changes the optimization parameter from algorithms to the positive real functions $\tilde{n}: \mathbb{R}^2 \to [0,\infty)$.
>
> - Lastly, we found out that the main problem happens in the 'instances with small suboptimality gaps' and made a novel modification (Opt2 -> Opt3), and use Jensen's inequality in a creative way (Claim 1 in Appendix D) to achieve a computable optimal answer.
>
> There are also other minor novel techniques, such as extension of the [Lemma 1, Kaufmann et al., 2016] (Lemma 11 in our paper) which helped us to transform the little-kl divergence to easier notation.
>
>
>
> > W5) The simulations are limited to just two sets of experiments. The baselines or compared algorithms are just 2/1 in the respective setting. And it might be better to include different values since the authors consider the fixed confidence setting and change priors as well.
>
> We made a brief comparison between Algorithm 1 and TTUCB in $K=10$ arm environment with different prior means, prior variances and instance variances. (because of the time constraint, it was not easy to finish TTTS at the same time.) Details of experiments are in the next comment.
>
> After 500 simulations, the expected stopping time of our algorithm was $1.1\times 10^5$, while TTUCB attains $7.1\times 10^5$. Even after giving an advantage to TTUCB, our algorithm shows a much smaller sample complexity than TTUCB.
>
> To avoid possible confusion, allow us to clarify our experiment result again. In Table 1, the compared algorithms have 10 times larger expected stopping time than ours, and about 500 times larger maximum stopping time.

---

> > ### Author Response · Authors · 2024-08-07
> > **Details on experiment (W5)**
> >
> > **Experiment Setting**
> >
> > - Number of arms $k=10$
> > - Prior means: we sample 10 random numbers before the experiment starts, and set them as prior means. Here's the list of prior means: [-0.053  0.528 -0.332 -0.368 -0.273  0.909 0.418 -1.17  0.873 -0.405]
> > - Prior variance: we sample 10 random numbers before the experiment starts, and set them as prior means. Here's the list of prior std: [0.604 1.477 1.163 0.988 0.560 0.513 0.997 1.332 0.828 0.833]
> > - Instance variance: we sample 10 random numbers before the experiment starts, and set them as prior means. Here's the list of instance std: [1.498, 1.262, 1.485, 0.963, 1.375, 0.969, 1.357, 1.238, 1.088, 0.699]
> > - Number of experiments: 500
> > - We stop additional sampling of TTUCB when its number of samples is over $10^8$ because of the time constraint. This means we gave some 'advantage' on TTUCB about expected stopping time (since it makes TTUCB stop earlier than it should.)

---

> > ### Comment · Reviewer_4HmL · 2024-08-07
> >
> > Thanks to the authors for addressing all my questions and concerns. The explanations are pretty convincing and I have bumped my score. Good luck.

---

### Official Review · Reviewer_bY8m · 2024-07-02

**Soundness:** 3
**Presentation:** 3
**Contribution:** 3
**Rating:** 6
**Confidence:** 4

**Summary:**

This paper considers a best arm identification problem in Bayesian multi-armed bandit setting. The arms' distributions are generated according to the unknown prior, and the probability of error is averaged over this prior distribution. The paper makes two key contributions: first, a lower bound characterizing the fundamental hardness of BAI in the Bayesian setting (where the L(H) parameter is identified as playing a key role), and second, and achievability part that proposes a successive elimination-based algorithm (with early stopping in case the arm instance has very close best arms). The algorithm achieves within a logarithmic factor of the lower bound.

**Strengths:**

The paper brings out an interesting twist on BAI, where the Bayesian setting is statistically easier on "an average" compared to the frequentist setting. The paper explains the intuition clearly throughout, and is executed competently.

**Weaknesses:**

There is no glaring weakness. It appears to be obvious in hindsight that "very close" instances do not occur with substantial probability, when we have a Gaussian prior, making the hard frequentist instances statistically irrelevant in the Bayesian setting. This makes the BAI problem easier in terms of the expected stopping time. If the prior model creates close instances with non-vanishing probability, I suppose we'd be back to the frequentist performance.

**Questions:**

The definition of $\mathcal H_\mu$ at the beginning of page 4 may not be precise...is it an event or a $\sigma$-algebra? In the former case, with continuous Gaussian priors, it is not clear if the event has non-zero probability, which makes all the conditional probabilities technically dubious. Please clarify.

**Limitations:**

Limitations are acknowledged in the conclusions reasonably.

---

> ### Author Rebuttal · Authors · 2024-08-05
>
> We thank the reviewer for the careful reading of the paper and the insightful comments. In the following, we address the question raised.
>
> > The definition of ${\mathcal{H}}_{\mu}$ at the beginning of page 4 may not be precise...is it an event or a $\sigma$-algebra? In the former case, with continuous Gaussian priors, it is not clear if the event has a non-zero probability, which makes all the conditional probabilities technically dubious. Please clarify.
>
> We are using the law of total expectation $\mathbb{E} [X] =  \mathbb{E} [\mathbb{E} [X|\sigma(Y)]] = \mathbb{E} [g(Y)]$ where $\sigma(Y)$ is the sigma algebra generated by random variable $Y$ and $g(y) = \mathbb{E} [X|Y=y]$. This argument holds even when {Y=y} has zero probability. See for example
>
> 1. Example 4.1.6 in 'Durrett, Rick. Probability: theory and examples. Vol. 49. Cambridge University Press, 2019'
> 2. Example 4 in https://www.stat.cmu.edu/~arinaldo/Teaching/36752/S18/Notes/lec_notes_6.pdf
> 3. https://math.stackexchange.com/questions/1332879/conditional-probability-combining-discrete-and-continous-random-variables
>
>  In particular, in our case $Y$ is $\mu$ and we are computing $\mathbb{E}\_{\mu_0 \sim \mathcal{H}} [\mathbb{P}(J\neq i^*(\mu)|\mu=\mu_0)]$. In this sense, $\mathcal{H}_{\mu_0} = \{\mu = \mu_0\}$ is an event. In the Bayesian setting, $\mu$ is a random variable, and we wanted to express the probability of error (PoE) as the form of the law of the total expectation, to emphasize that our probability of error is the marginalized form of the PoE. We used the notation $\mathcal{H}\_{\mu}$ to simplify notation.

---

> > ### Comment · Reviewer_bY8m · 2024-08-13
> > **Thank you for your response.**
> >
> > I appreciate your technical clarification. I keep my score.

---

### Official Review · Reviewer_EP71 · 2024-07-09

**Soundness:** 3
**Presentation:** 2
**Contribution:** 3
**Rating:** 6
**Confidence:** 3

**Summary:**

In this paper, the focus is on the fixed-confidence best-arm identification problem within a Bayesian setting. The objective is to determine the arm with the largest mean with a certain confidence, given a known prior distribution. Existing work in FC-BAI is mostly in the frequentist setting. This paper shows that the popular algorithms used in the frequentist setting are not optimal in this setting. Additionally, a lower bound for this setting is derived, using a sampling complexity constant and the confidence level. Lastly, a successive elimination method that matches the order of the derived lower bound along with experiments comparing the performance of popular frequentist algorithms and this algorithm are presented.

**Strengths:**

- The authors show that popular frequentist algorithms are suboptimal in the Bayesian setting.

- The paper presents a novel algorithm, which is significant because, according to related work, it is uncommon to have a stopping time in Bayesian optimization.

- In the successive elimination method, the indifference zone is not a parameter.

- The derived lower bound is novel.

**Weaknesses:**

1. 47: Example 1 lacks proof or reference for the number of samples.

2. The authors do not discuss how they derived the constant L(H) and its significance. Since this constant represents sampling complexity and is not a traditional term, it is harder to compare the complexity to existing work.

3. 171: In Chapter 3, the authors claim there is a known lower bound for the expected stopping time without providing any reference.

4. 196: The derivation of the inequalities in Section 3.1 is not clear.

5. 242-243: In Chapter 5, there is no explanation for why this particular confidence width was considered.

6. In Chapter 6, there is no discussion on why Algorithm 1 shows higher error percentages than the other two algorithms.

7. 313: there are two dots instead of one.

**Questions:**

- 267-268: In Remark 3, can the authors explain if in the general case the result matches with the lower bound?
- 309-313: The authors state that the expected stopping time of the track-and-stop method is at least half of the TTS and TTUCB methods, which according to Table 1, is on average at least 300. This number is less than that for Algorithm 1, so can the authors explain why they didn’t compare?

**Limitations:**

The research was done only for the Gaussian distribution.

---

> ### Author Rebuttal · Authors · 2024-08-05
>
> We thank the reviewer for the careful reading of the paper and the insightful comments. We will revise the typos and references in our final version. In the following, we address the main questions raised.
>
> > W2) The authors do not discuss how they derived the constant L(H) and its significance. Since this constant represents sampling complexity and is not a traditional term, it is harder to compare the complexity to existing work.
>
> From Section 3, we find that measuring the probability when $\mu$ achieves a small suboptimality gap is crucial. The constant $L(H)$ represents the ratio between the gap $\Delta$ and the probability that $\mu$ achieves a suboptimality gap smaller than $\Delta$, as outlined in Lemma 1. We will include a discussion on this in our final version.
>
> > Q1) 267-268: In Remark 3, can the authors explain if in the general case, the result matches with the lower bound?
>
> Our lower bound holds only when $\delta$ is sufficiently small, whereas the result in Remark 3 (upper bound) is the case of moderately large $\delta$, so we cannot say the result matches with the lower bound.
>
> > Q2) 309-313: The authors state that the expected stopping time of the track-and-stop method is at least half of the TTTS and TTUCB methods, which according to Table 1, is on average at least 300. This number is less than that for Algorithm 1, so can the authors explain why they didn’t compare?
>
> To avoid possible confusion, allow us to clarify our experiment result again. In Table 1 of the paper we submitted, our algorithm has a stopping time of $10^4$, TTUCB has a stopping time of $1.5\times 10^5$, and TTTS has an even higher stopping time. Even if the stopping time of Track-and-Stop is half that of TTUCB, it would be $7.6\times 10^4$, which is still over 7 times larger than our algorithm.
>
> > W4) 196: The derivation of the inequalities in Section 3.1 is not clear.
>
> We will add more explanation on Section 3.1 in our final version. Corollary 3 implies that for any fixed mean vector $\mu \in \mathbb{R}^2$, the stopping time is lower bounded by $\log (1/\delta)/(\mu_1 -\mu_2)^2$. For the following sequence of equations (equation in line 196~197),
>
> $\mathbb{E}\_{\mu \sim H} [\tau] \geq \mathbb{E}\_{\mu \sim H} [\tau \cdot  \mathbb{1}[|\mu_1 -\mu_2| \leq \epsilon]]
>     \geq \frac{\log \delta^{-1}}{\epsilon^2} \mathbb{P}\_{\mu \sim H} [|\mu_1- \mu_2| \leq \epsilon] = \Omega(\frac{\log \delta^{-1}}{\epsilon})$
>
> 1) The first inequality holds since $\tau$ is a positive random variable.
> 2) The second inequality holds by the following
>   - law of total expectation $\mathbb{E}\_{\mu \sim H} [\tau \cdot  \mathbb{1}[|\mu_1 -\mu_2| \leq \epsilon]] = \mathbb{E}\_{\mu \sim H} [\mathbb{E}[\tau |\mu] \cdot  \mathbb{1}[|\mu_1 -\mu_2| \leq \epsilon]]$
>   - and Corollary 3 $\mathbb{E}\_{\mu \sim H} [\mathbb{E}[\tau|\mu] \cdot  \mathbb{1}[|\mu_1 -\mu_2| \leq \epsilon]] \geq \mathbb{E}\_{\mu \sim H} [\frac{\log \delta^{-1}}{(\mu_1 - \mu_2)^2} \cdot  \mathbb{1}[|\mu_1 -\mu_2| \leq \epsilon]]\geq \frac{\log \delta^{-1}}{\epsilon^2} \mathbb{P}\_{\mu \sim H} [|\mu_1- \mu_2| \leq \epsilon]$
> 3) Finally, by Lemma 1, we can replace $\mathbb{P}(\cdots)$ as $L(H)\cdot \epsilon$ which leads the RHS.
>
> > W5) 242-243: In Chapter 5, there is no explanation for why this particular confidence width was considered.
>
> The bound is for assuring $LCB(i,t) \leq \mu_i \leq UCB(i,t)$ for all $t\in \mathbb{N}$ and $i\in[k]$ with high probability. For more details, please check Lemma 15 (Appendix E). In our final version, we will add more explanation about the proof in Section 5.
>
> > W6) In Chapter 6, there is no discussion on why Algorithm 1 shows higher error percentages than the other two algorithms.
>
> When our algorithm finds the suboptimality gap is too narrow, instead of trying to identify the best arm, our algorithm stops sampling to avoid excessive sample complexity. On the other hand, the other two algorithms always keep sampling until they identify the best arm. This difference causes the error rate difference. However, as we have mentioned in Section 3, the algorithm should set an indifference condition in the Bayesian setting or the algorithm will attain an excessive scale of sample complexity, even diverging to infinity in expectation.

---

> > ### Comment · Reviewer_EP71 · 2024-08-09
> >
> > Thank you for providing detailed explanations in response to the points I raised in my review. I appreciate the effort and clarity with which you addressed my concerns.
> > After reviewing your responses, I understand and acknowledge the explanations provided. While your rebuttal has clarified several aspects of your work, I believe that a score of 6 is an appropriate reflection of the significance of your research. At this point, I do not plan on changing my score.
> > I wish you the best of luck with your submission.

---

### Official Review · Reviewer_jd9J · 2024-07-10

**Soundness:** 4
**Presentation:** 4
**Contribution:** 3
**Rating:** 7
**Confidence:** 5

**Summary:**

The paper studies the problem of FC-BAI; the goal is to find the arm with largest mean with a given probability of correctness. It analyzes FC-BAI for a Gaussian bandit model where the arms reward mean is sampled from a known prior and the reward variances are known. It proves other (frequentist) FC-BAI algorithms could fail at this setting. Then it proves a lower bound for the sample complexity of this problem. Finally, an elimination algorithm is introduced which is near-optimal. The main difference of this algorithm with previous algorithms is in its less conservative stopping criteria, where it stops if the remaining arms have small enough optimality gap.

**Strengths:**

*Soundness*: the paper clearly defines the problem, introduces a lower bound for the problem, theoretically proves the sub-optimality of previous algorithms, and conducts simulation studies along with ablation experiment.

*Presentation*: The paper provides all the proof sketches and pseudo-codes required for reproducibility of the results. It efficiently uses its notation and avoids overloading symbols.

**Weaknesses:**

*Contribution*: The paper considers a small subset of bandit models, Gaussian reward with know prior and variance. This limits applicability of the algorithm and theoretical results. As also mentioned in the discussion section, it would be great if they could generalize the algorithm to deal with misspecified or under-specified prior (see [1]), or unknown variance.


[1] Azizi, J., Kveton, B., Ghavamzadeh, M. and Katariya, S. 2023. Meta-Learning for Simple Regret Minimization. Proceedings of the AAAI Conference on Artificial Intelligence. 37, 6 (Jun. 2023), 6709-6717. DOI:https://doi.org/10.1609/aaai.v37i6.25823.

**Questions:**

1. In Thm 4, the constants in the lower bound seem very small ($\frac{1}{16 e^4} \approx \frac{1}{2^8}$), so it could entail a trivial lower bound in most cases, specially since it is the lower bound to the sample complexity (natural number). What is the actual value of this lower bound in some of your example simulations?

**Limitations:**

Yes.

---

> ### Author Rebuttal · Authors · 2024-08-05
>
> We are truly grateful for the encouraging evaluation and valuable comments. The following is the response to the question you raised.
>
> >In Thm 4, the constants in the lower bound seem very small ($\frac{1}{16e^4}\approx 2^{-8}$), so it could entail a trivial lower bound in most cases, especially since it is the lower bound to the sample complexity (natural number). What is the actual value of this lower bound in some of your example simulations?
>
> We agree that our lower bound tends to be much smaller than actual values. For example, when $\delta=0.01$, for the instance in our Example 1 ($\mu=(1,0.9,0.1)$), the sample complexity lower bound is 0.0468, smaller than 1. However, the main objective of our lower bound is to check the order of the sample complexity. Through this analysis, we found out that L(H) is a crucial variable in designing the algorithm, and we were able to propose our Algorithm 1 which is optimal up to a logarithmic factor.

---

### Decision · Program_Chairs · 2024-09-25

**Decision:**

Accept (poster)

**Comment:**

This paper considers a Bayesian formulation of the classic fixed confidence best-arm identification problem. In particular, they consider the special case of known independent Gaussian priors for each arm, and Gaussian rewards with known variance. Given the overlapping support of the priors, there is a non-zero chance of an arbitrarily small minimum gap which blows up the sample complexity of naively applied delta-PAC algorithms for the frequentist setting. The main contributions are a novel lower bound for this setting and an algorithm that matches it up to log factors.

The goals of the paper are clear and they are well-executed. Perhaps the biggest weakness of this paper is the assumption that the prior distributions are Gaussian. As the critical identified quantity in the sample complexity denoted as L(H) is derived directly from this assumption, it is unlikely to be meaningful in the general prior case. Indeed, if the prior is a uniformly random permutation of a fixed set of separated means, L(H) has no impact on the sample complexity of this problem (see Chen, Li, Qiao 2016 and Simchowitz and Jamieson 2016). As some reviewers suggested, the authors would improve the paper by adding a longer discussion motivating L(H) and under what prior conditions it is a relevant quantity.